https://doi.org/10.1038/s41467-022-31054-4　　**OPEN**

# Ly6D$^+$Siglec-H$^+$ precursors contribute to conventional dendritic cells via a Zbtb46$^+$Ly6D$^+$ intermediary stage

Konstantin Lutz [1,12], Andrea Musumeci [1,12], Christopher Sie [2], Ezgi Dursun[1], Elena Winheim [1], Johannes Bagnoli[3], Christoph Ziegenhain [3,4], Lisa Rausch[1], Volker Bergen [5,6], Malte D. Luecken [5], Robert A. J. Oostendorp [7], Barbara U. Schraml [8,9], Fabian J. Theis [5,6], Wolfgang Enard [3], Thomas Korn [2,10,11] & Anne B. Krug [1✉]

Plasmacytoid and conventional dendritic cells (pDC and cDC) are generated from progenitor cells in the bone marrow and commitment to pDCs or cDC subtypes may occur in earlier and later progenitor stages. Cells within the CD11c$^+$MHCII$^{-/lo}$Siglec-H$^+$CCR9$^{lo}$ DC precursor fraction of the mouse bone marrow generate both pDCs and cDCs. Here we investigate the heterogeneity and commitment of subsets in this compartment by single-cell transcriptomics and high-dimensional flow cytometry combined with cell fate analysis: Within the CD11c$^+$MHCII$^{-/lo}$Siglec-H$^+$CCR9$^{lo}$ DC precursor pool cells expressing high levels of Ly6D and lacking expression of transcription factor Zbtb46 contain CCR9$^{lo}$B220$^{hi}$ immediate pDC precursors and CCR9$^{lo}$B220$^{lo}$ (lo-lo) cells which still generate pDCs and cDCs in vitro and in vivo under steady state conditions. cDC-primed cells within the Ly6D$^{hi}$Zbtb46$^-$ lo-lo precursors rapidly upregulate Zbtb46 and pass through a Zbtb46$^+$Ly6D$^+$ intermediate stage before acquiring cDC phenotype after cell division. Type I IFN stimulation limits cDC and promotes pDC output from this precursor fraction by arresting cDC-primed cells in the Zbtb46$^+$Ly6D$^+$ stage preventing their expansion and differentiation into cDCs. Modulation of pDC versus cDC output from precursors by external factors may allow for adaptation of DC subset composition at later differentiation stages.

[1] Institute for Immunology, Biomedical Center, LMU Munich, Großhaderner Str. 9, 82152 Planegg-Martinsried, Germany. [2] Institute for Experimental Neuroimmunology, Technical University of Munich, School of Medicine, Ismaninger Str. 22, 81675 Munich, Germany. [3] Anthropology and Human Genomics, Faculty of Biology, LMU Munich, Großhaderner Str. 2, 82152 Planegg-Martinsried, Germany. [4] Department of Cell and Molecular Biology, Karolinska Institute, Stockholm, Sweden. [5] Institute of Computational Biology, Helmholtz Center Munich, Ingolstädter Landstr. 1, 85764 Neuherberg, Germany. [6] Department of Mathematics, Technical University of Munich, Boltzmannstraße 3, 85748 Garching, Germany. [7] Technical University of Munich, School of Medicine, Department of Internal Medicine III, Ismaninger Str. 22, 81675 Munich, Germany. [8] Institute for Cardiovascular Physiology and Pathophysiology, Biomedical Center, LMU Munich, Großhaderner Str. 9, 82152 Planegg-Martinsried, Germany. [9] Walter-Brendel-Centre of Experimental Medicine, University Hospital, LMU Munich, 82152 Planegg-Martinsried, Germany. [10] Department of Neurology, Technical University of Munich School of Medicine, Ismaninger Str. 22, 81675 Munich, Germany. [11] Munich Cluster for Systems Neurology (SyNergy), Feodor-Lynen-Str. 17, 81377 Munich, Germany. [12] These authors contributed equally: Konstantin Lutz, Andrea Musumeci. ✉email: anne.krug@med.uni-muenchen.de

Dendritic cells (DC) are critical in the defense against pathogens and the efficacy of vaccines. As highly efficient antigen-presenting cells and producers of polarizing cytokines, DCs induce protective long-lasting T cell and humoral immune responses. DC subpopulations are conserved across species[1]. Their specific functional properties are related to ontogeny and distinct transcriptional programs[2]. In addition, environmental cues in lymphoid and non-lymphoid tissue niches shape DC functionality[3,4]. While conventional DC 1 (cDC1) efficiently promote cytotoxic T cell and Th1 responses, cDC2 are superior in inducing T helper (Th1, Th2, and Th17) responses[5]. Plasmacytoid DCs (pDC) are unique cells with a plasma-cell like morphology circulating in the blood and residing in lymphoid organs, which participate in the antiviral defense by rapid and massive production of type I interferons (IFNs) and by promoting natural killer (NK) cell and T cell responses[6–11]. Due to their extraordinary ability to produce IFN I, pDCs are important for innate antiviral defense but they also promote systemic autoimmunity[10–15]. Being short-lived and non-proliferative, pDCs are continuously replenished by progenitor cells in the bone marrow (BM)[16]. cDCs and pDCs were thought to be derived from common DC progenitors (CDP), which give rise to cDC-committed precursors (pre-cDC) and pDCs in a Flt3 ligand (Flt3L) dependent manner[17–19]. CDPs and pre-cDCs in the BM were found to be heterogeneous and contain cells that are already primed for differentiation into a specific DC subpopulation, suggesting commitment may occur continuously at earlier and later developmental stages starting with lymphoid primed multipotent progenitors (LMPPs) or even hematopoietic stem and progenitor cells (HSPC)[20–27]. It was shown that both myeloid and lymphoid progenitors can give rise to pDCs as well as cDCs in the presence of Flt3L[28,29]. Still, fate-mapping experiments in mice have shown that lymphoid progenitors (LPs) generate cDCs in neonates, but do not make a substantial contribution to the cDC pool in adult mice[4,30–32].

Specification and maintenance of pDCs require expression of E protein transcription factor E2-2 (*Tcf4*)[33,34]. Upon loss of E2-2, pDCs acquire a cDC-like phenotype[34,35]. E2-2 is counteracted by E-protein inhibitor ID2, which promotes cDC development and is repressed in developing pDCs[36,37], suggesting that pDCs and cDCs are developmentally linked[38]. This view was challenged by recent studies, which identified CD11c−Ly6D+Siglec-H+ pDC-committed precursor cells in mouse BM[39,40]. These were shown to derive from IL7R+ lymphoid progenitors (LP), which also give rise to B lymphocytes although a common clonal progenitor of B cells and pDCs has not been described[39–41]. The contribution of myeloid progenitors and CDPs to the pDC lineage has been questioned as bona fide pDCs were shown to develop from IL7R+Ly6D+ LPs whereas CD115+Ly6D− CDPs generated exclusively cDCs[39]. These findings suggested that pDC and cDC develop independently in separate lineages after the lymphoid-primed multipotent progenitor (LMPP) stage. However, pDC vs. cDC (especially cDC1) cell fate was shown to be regulated by the antagonistic transcription factors (TFs) E2-2 (*Tcf4*) and ID2[34,36,38]. This cross-regulation between pDC and cDC differentiation is inconsistent with completely separate pDC and cDC lineages and suggests a developmental relationship[38].

CD11c+ DC precursors with a pDC-like phenotype in mouse BM, which gave rise to both pDCs and cDCs in vitro and after transfer in vivo were previously identified[42,43]. These cells were characterized by expression of CD11c and pDC surface markers such as Siglec-H and BST2, but showed absent or low expression of MHC class II and CCR9. In contrast, differentiated pDCs are MHCII+ and express high levels of CCR9[43]. These pDC-like cells responded to endosomal TLR stimulation with type I IFN production like pDCs, but failed to present antigens on MHC class II

after antigen targeting in vivo in contrast to CCR9hi differentiated pDCs[44]. Studies characterizing the pre-DC compartment in mouse BM by single cell RNA-sequencing (scRNAseq) showed that CD11c+Siglec-H+ pre-DCs were heterogeneous and generated both cDCs and pDCs in vitro and in vivo[26] with Siglec-H+ Ly6C− pre-DCs being enriched for pDC-primed cells[39]. pDC-like cells expressing Siglec-H, Zbtb46 and CX3CR1 were also found in mouse spleen[40,45] and share many features of Axl+Siglec-6+ DC (also designated as pre-DC) in human blood[46–48]. It is still debated if these are "transitional"[45] or "non-canonical"[38] DCs with specific functions or immediate precursors of cDCs[46].

Here we investigate the heterogeneity and commitment of DC precursor subsets using single-cell RNA sequencing and trajectory inference from RNA velocity combined with cell fate analysis. Within the CD11c+Siglec-H+ precursor compartment Ly6D− cells are cDC-committed and Ly6DhiZbtb46−CCR9lo B220hi cells are immediate precursors of pDCs. Ly6D-hiZbtb46−CCR9loB220lo cells gave rise to pDCs by upregulating B220 and CCR9, but are still capable of generating cDCs via a Zbtb46+Ly6D+ intermediary stage, which can be modulated by type I IFN.

## Results

**Relationship of CD11c+Siglec-H+CCR9loB220hi cells to pDCs.** Since CD11c+Siglec-H+CCR9lo precursors in mouse BM were shown to give rise to both cDCs and pDCs[42,43], we revisited heterogeneity in this compartment using multiparameter flow cytometry. We observed heterogeneous expression of B220 within the CD11c+Siglec-H+CCR9lo BM cell fraction (Fig. 1a). Higher expression of B220 coincided with higher levels of Siglec-H, BST2, Sirp-α, MHCII, Ly6C, Ly6D, Ly49Q, CCR4, and CCR5 surface expression, whereas lower B220 expression within the CD11c+Siglec-H+CCR9lo precursor fraction correlated with higher expression of CD135 (Flt3), CD115 (MCSFR) and CXCR4 (Supplementary Fig. 1). Using bulk RNA-sequencing we compared the transcriptome of CCR9loB220lo and CCR9loB220hi cells within the CD11c+Siglec-H+CCR9lo precursor pool (called lo-lo and lo-hi cells from now on) as well as Siglec-H− pre-cDCs and differentiated CCR9hiB220hi pDCs sorted from BM cells (Fig. 1a). Principal component analysis showed the close relationship of the lo-hi precursor cells to pDCs whereas lo-lo precursor cells were more similar to Siglec-H− pre-cDCs (Fig. 1b). Hierarchical clustering of 2880 genes differentially expressed (DEGs) between the four populations led to 13 clusters of coregulated genes (Supplementary Fig. 2). Several clusters contained genes expressed at higher levels in both pDCs and lo-hi precursors, including the pDC signature genes *Siglech*, *Bst2*, *Tcf4* (E2-2), *Spib*, *Bcl11a*, *Sell* (CD62L), *Klra17* (Ly49Q), *Ly6d* and *Runx2*. Within these clusters, several genes including genes encoding MHC II subunits, *Ciita*, *Cd74*, several interferon-stimulated genes (*Ifit2*, *Ifi213*, *Mx1*), *Irf7*, *Jak1*, *Relb*, *Ccr9*, *Ccr5*, and *Cd8a*, showed higher expression in pDCs than in lo-hi precursors indicating a more differentiated state. Two major clusters contained genes expressed at higher levels in pre-cDCs and lo-lo precursors. Several clusters contained genes expressed at higher levels in pre-cDCs than in the other populations. These included genes typically expressed in the cDC lineage, e.g., *Zbtb46*, *Csf1r*, *Cx3cr1*, *Spi1*, *Batf3*, *Irf4*, *Id2*, and *Cd40* (Supplementary Fig. 2).

Genes involved in cell adhesion, antigen processing and presentation, chemokine signaling, Fc-gamma receptor-mediated phagocytosis and MAPK signaling and genes involved in primary immunodeficiency, Jak-Stat signaling, and Erbb signaling pathways were overrepresented in the clusters that showed higher expression in pDCs and lo-hi cells (Supplementary Fig. 2 and Supplementary Data 1). Genes with binding sites for TFs Stat1/

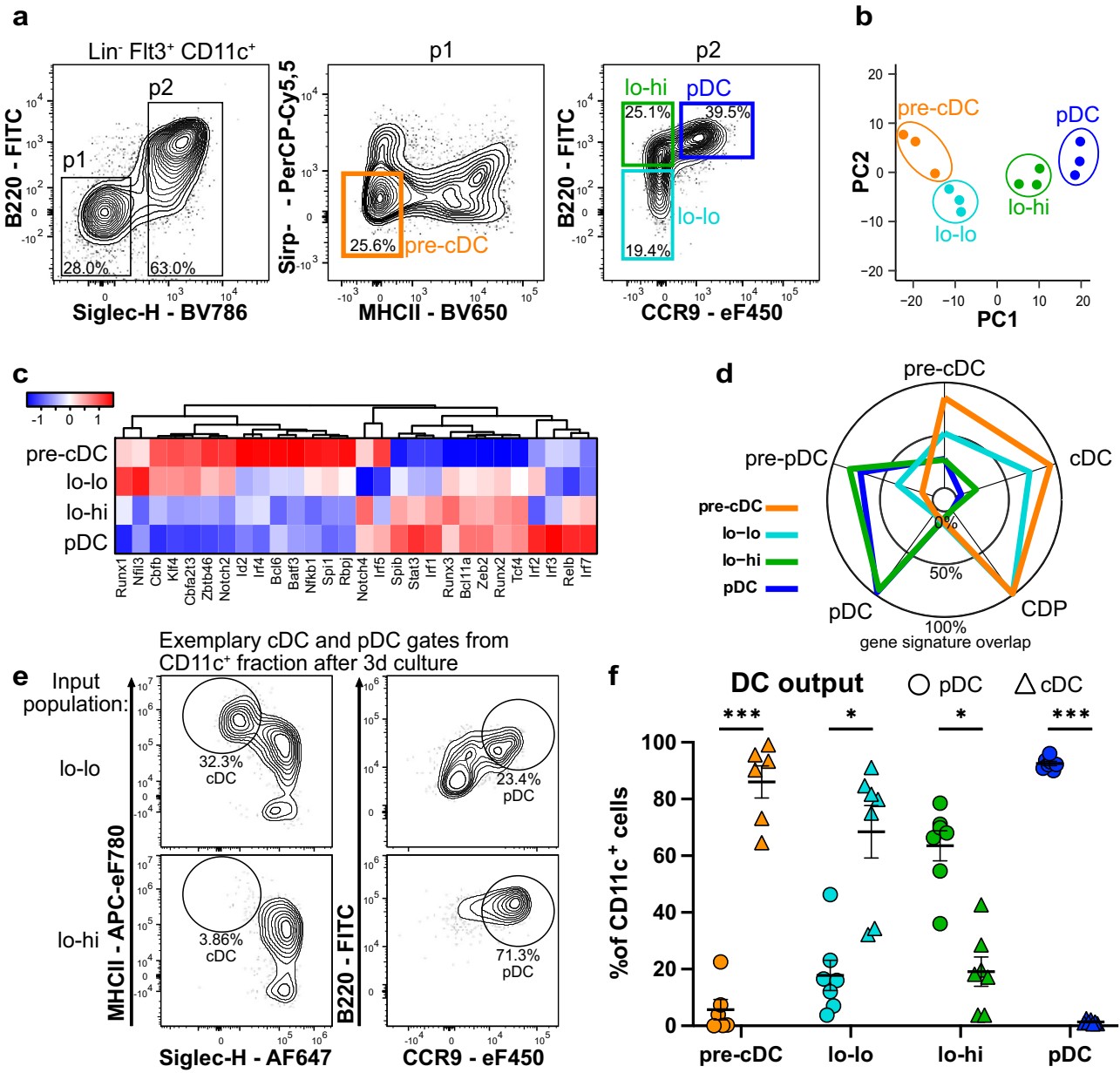

**Fig. 1 Relationship of Siglec-H⁺CCR9^loB220^hi cells and pDCs. a** Gating strategy for Siglec-H⁻ pre-cDC, Siglec-H⁺CCR9^loB220^lo (lo-lo), CCR9^loB220^hi (lo-hi) and CCR9^hiB220^hi pDCs. The indicated populations were sorted from the BM cells of 3 individual C57BL/6 mice and processed for RNA-sequencing. **b** Principal component analysis performed on normalized counts using all genes. **c** Expression heatmap of selected TFs, z-score hierarchical clustering by Euclidean distances (normalized counts). **d** Gene signatures of the sorted populations compared to previously published gene signatures of pre-cDC, cDC, CDP, pDC and pre-pDC[39]. Distance from the middle of the radar plot indicates the percentage of overlap between genes of sorted populations that showed higher expression than the inter-population median and the indicated gene signatures. **e** Gating strategy for pDC/cDC output after culturing precursor cells. cDCs were identified as Siglec-H⁻ MHCII^hi cells and pDCs as Siglec-H⁺CCR9^hiB220^hi cells within CD45⁺CD11c⁺ cells. **f** Siglec-H⁻ pre-cDC, lo-lo and lo-hi precursors, and pDC were sorted from Lineage-depleted BM cells of WT mice as shown in (**a**) and cultured for 3 days on EL08-1D2 stromal cells in Flt3L-containing medium. The percentages of cells with a phenotype of pDC (circles) or cDC (triangles) within CD11c⁺ progeny of the indicated input populations is shown as mean ± SEM (n = 7) and was compared using paired, two-sided t-tests with Holm-Šídák correction for multiple testing. Adjusted p-values: <0.05(*), <0.005(**), <0.0001(***).

Stat2, Irf1/2/3/7, and Irf8/9 and several targets of these TFs were found to be enriched here and were most highly expressed by pDCs including *Irf1*, *Irf7*, *Tlr7*, *Stat1*, *Stat2*, *Stat3*, *Bst2*, *Cd4*, and *Kdr* (encoding Vegfr-2). The major cluster with higher expression in pre-cDCs and lo-lo cells was enriched for genes involved in DNA replication, cell cycle, pyrimidine/purine metabolism and p53 signaling with binding motifs for TFs of the E2F family and Rb1 consistent with a proliferative precursor state. The cluster of genes with highest expression in pre-cDCs was enriched for genes

involved in Fc-gamma receptor-mediated phagocytosis, hematopoietic cell lineage, cytokine-cytokine receptor interaction, MAPK signaling and TLR signaling, which have Spi1/Spib/Spic and Irf4 binding motifs (Supplementary Fig. 2 and Supplementary Data 2).

As shown in Fig. 1c, lo-lo precursors expressed several TFs involved in the generation and function of cDC subsets at higher levels than lo-hi cells and pDCs. In contrast, lo-hi precursors showed a higher expression of TFs involved in pDC generation

and function compared to the lo-lo precursors and pre-cDCs. Expression of *Irf2, 3, 7* and *Relb* appeared to be even higher in differentiated pDCs. Comparison of the gene expression profile with published gene expression signatures (Fig. 1d) confirmed the closer relationship of the lo-lo fraction with pre-cDCs, cDCs, and CDPs and of the lo-hi fraction with pDCs and the recently described pre-pDCs[39]. Thus, comparative transcriptome analysis suggested lo-hi cells to be advanced precursors of pDCs with lower expression of genes indicating proliferation compared to pre-cDC and lo-lo cells. They shared the gene expression profile of pDCs to a large extent but expressed lower levels of MHC II and genes involved in the induction of type I IFNs and IFN-response than fully differentiated pDCs, consistent with an immediate precursor state.

To investigate the cell fate of the lo-hi precursors in comparison to the lo-lo precursors, pre-cDCs and pDCs, these populations were sorted from BM cells as shown in Fig. 1a and their phenotype was analyzed after 3 days of culture with feeder cells and Flt3L. The lo-lo precursors generated more cDCs than pDCs. Cells with lo-hi phenotype were also detected in the progeny of lo-lo precursors indicating transition from the lo-lo to the lo-hi stage (Fig. 1e). The lo-hi precursors mainly gave rise to CCR9[hi] pDCs and fewer cells with cDC phenotype (Fig. 1e, f). Siglec-H[−] pre-cDCs generated cDCs as previously reported, and CCR9[hi] pDCs largely maintained their pDC phenotype. These results are in line with the results of global gene expression analysis and indicate that CD11c[+]Siglec-H[+]CCR9[lo] BM cells expressing high levels of B220 are biased to differentiate into CCR9[hi] pDCs while those expressing low levels of B220 are biased to generate cDCs.

**Transcriptional dynamics derived from single-cell RNA sequencing analysis indicate cell fate transitions in the DC precursor compartment.** Ly6D was recently identified as a marker of pDC-committed progenitors described as CD11c[−]IL7R[+]Ly6D[+] Siglec-H[+] LP[39,40]. We found that lo-hi precursors were also uniformly Ly6D[hi] (Supplementary Fig. 1). A fraction of Ly6D[+] cells was also found in the lo-lo precursors (Supplementary Fig. 1). These data suggested that Ly6D could also mark pDC-committed precursors contained within the CD11c[+]Siglec-H[+]CCR9[lo] compartment. To further investigate the relation of lo-lo and lo-hi precursors to the Ly6D[+] pDC progenitors[39,40] and other progenitor populations in the mouse BM, we performed scRNA-seq using mcSCRB-seq[49]. CD115[+] CDP, CLP, Ly6D[+]Siglec-H[−] LP, Ly6D[+]Siglec-H[+] LP (defined by Rodrigues et al.[40]), and 3 CD11c[+] precursor fractions (Siglec-H[−] pre-cDC, lo-lo, lo-hi), as well as pDCs, were single-cell sorted from Lineage-depleted wildtype BM cells (Supplementary Fig. 3) into plates for scRNA-seq. 675 cells retained after filtering were visualized in a two-dimensional diffusion map with pseudo-temporal ordering of cells[50] based on expression of the top 10,000 highly variable genes (Fig. 2a). Ly6D[+]Siglec-H[+] LP were projected between Ly6D[+]Siglec-H[−] LP and lo-hi precursors (Fig. 2a). Cells sorted as CLP overlapped with both Ly6D[+] LP populations as expected. Cells sorted as lo-hi precursors were projected between Ly6D[+]Siglec-H[+] precursors and pDCs in a continuous distribution, consistent with an immediate pDC precursor state that follows the Ly6D[+]Siglec-H[+] pDC progenitor stage. CDP and Siglec-H[−] pre-cDC clustered together distant from pDC as expected. The lo-lo precursor cells were heterogeneous. One fraction expressing high levels of *Ly6d* projected close to lo-hi cells and one fraction not expressing *Ly6d* clustered with pre-cDCs (Fig. 2a, b) suggesting that lo-lo precursors still contain Ly6D[−]Siglec-H[+] pre-cDCs. Indeed, flow cytometry confirmed that a subset of cells within the lo-lo precursors expressed Zbtb46 a marker of cDC-committed precursors[51] (Fig. 2c).

Louvain community detection identified seven clusters, which were annotated by DEG and marker gene expression (Fig. 2d, f and Supplementary Fig. 4c). Cl. 0 contained cells sorted as CDP and pre-cDC marked by higher expression of genes typically found to be expressed in these cells (e.g., *Csf1r, Lgals3, Cd209a, Id2, Batf3, Ly6c2*). Cl. 6 cells sorted as CDP showed additional expression of *Mpo, Irf2* and *Cd34* consistent with an earlier myeloid progenitor phenotype. Cl. 2 contained cells sorted as Ly6D[+]Siglec-H[−] LP and CLP which expressed genes identifying them as B cell precursors (e.g., *Ebf1, Dntt, Vpreb3, Pax5*, and *Cd79a*). These also showed the highest expression of *Flt3* and *Cd81* compared to the other clusters. Cells sorted as CLP, Ly6D[+] Siglec-H[+] LP and lo-hi precursors occupied cl. 5, 4 and 3, with increasing expression of pDC marker genes *Bst2, Siglech, Tcf4, Spib* as well as *Ifnar1*. Cl. 1 denotes finally differentiated pDCs indicated by high expression of pDC marker genes and genes indicating final differentiation (e.g., *Ccr9, Tlr7, Cd8a, Cd74,* and MHC II genes Fig. 2f, Supplementary Fig. 4c).

To gain further insights into the developmental trajectories we performed RNA velocity analysis. We used scVelo, a likelihood-based dynamical model, to investigate the transcriptional dynamics of gene-specific splicing kinetics and infer RNA velocities[52]. The combination of RNA velocities across genes was used to calculate cell-to-cell transition probabilities. The RNA velocity vector field was projected onto the diffusion map with Louvain clusters indicated (Fig. 2d and Supplementary Fig. 4a). RNA velocity vectors in CDP/pre-cDC (cl. 0) pointed away from the other clusters. *Ms4a6c, Emilin2, Tnfaip2* (A20) and *Anxa1* were among the top 30 dynamics-driving genes in this cluster identified via scVelo-based likelihoods (see individual velocity plots in Supplementary Fig. 4b). *Csf1r, Batf3* and *Id2* also showed positive velocity in this cluster indicating upregulation in CDP and pre-cDC (Fig. 2g and Supplementary Fig. 4b). Vectors in cl. 6 (myeloid progenitors) and cl. 5 (Ly6D[+]Siglec-H[+] LP and CLP) pointed to cl. 0 (CDP/pre-cDC) indicating that cells in both of these clusters may give rise to CDPs and/or pre-cDCs (Fig. 2d and Supplementary Fig. 4a). RNA velocity in these clusters was dominated by genes involved in proliferation and survival in line with their progenitor function (e.g., *Top2a, Casc5, Kif11, Cenpe,* Supplementary Fig. 4b). In cl. 2, containing a fraction of cells sorted as Ly6D[+]Siglec-H[−] LP, RNA velocity vectors pointed towards more advanced B cell precursors or towards Ly6D[+] Siglec-H[+] LP in cl. 5 and 4. Several genes involved in B cell development showed positive RNA velocity and expression (e.g., *Pax5, Ebf1, Gpr97, Lax1, Xrcc6* (Ku70)) in B cell precursors indicating upregulation (Fig. 2g and Supplementary Fig. 4b). *Flt3* though highly expressed in B cell precursors showed negative RNA velocity indicating downregulation (Fig. 2g), which is consistent with the important function of Pax5-mediated Flt3 repression during B cell development[53]. The majority of cells in cl. 4 containing mainly Ly6D[+]Siglec-H[+] LP and lo-lo cells showed RNA velocity vectors pointing towards cl. 3. Cl. 4 cells showed for example positive RNA velocity for *Tcf4* and *Siglech* (Fig. 2g) indicating their upregulation in earlier Ly6D[+]Siglec-H[+] pDC progenitors. RNA velocity vectors in cl. 3 containing some Ly6D[+]Siglec-H[+] LP, but mainly lo-hi precursors pointed towards differentiated pDCs. These RNA velocity vectors were dominated by genes highly expressed in fully differentiated pDCs including *Klra17* (Ly49Q), *Ccr9, Rnaset2b, Siglech, Clec10a,* and *Tcf4* (Fig. 2g and Supplementary Fig. 4b) further supporting that these cells are immediate precursors of pDCs. These results are consistent with Ly6D[+]Siglec-H[−] LP containing mainly B cell precursors and Ly6D[+]Siglec-H[+] LP containing mainly pDC-committed cells[40] that further differentiate via Ly6D[+] lo-lo and lo-hi precursors to pDCs.

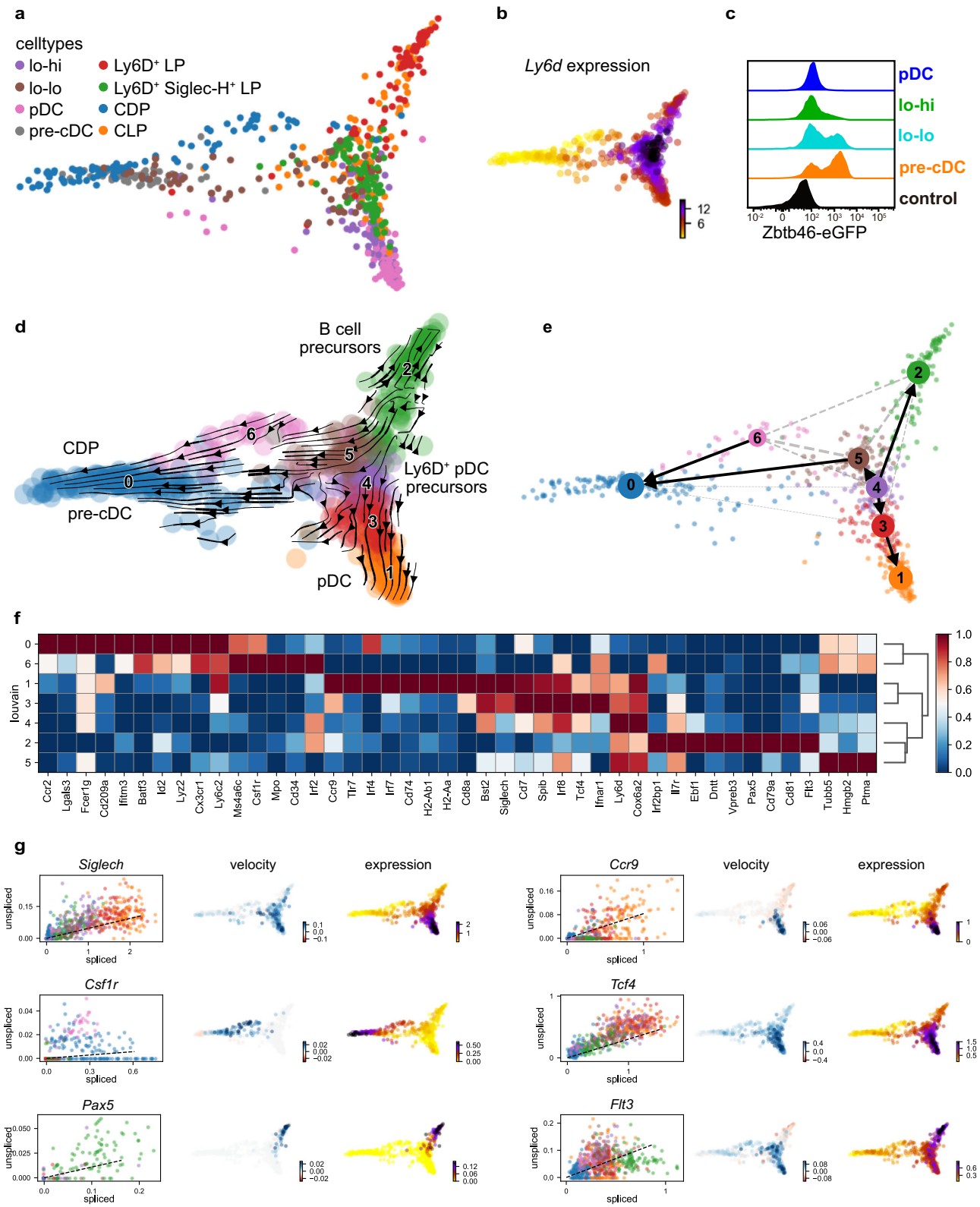

We next inferred trajectories between clusters using partition-based graph abstraction (PAGA) with directionality deduced from RNA velocity which uses a velocity-inferred pseudotime as prior (Fig. 2e). This analysis confirmed the continuous transition from Ly6D+Siglec-H+ LP and the Ly6D+ part of the lo-lo precursors (cl. 4) to lo-hi precursors (cl. 3) to pDCs (cl. 1). Cells in cl. 4 were also closely connected with B cell precursors (cl. 2).

Interestingly, PAGA also identified trajectories from cl. 4 to cl. 5 and from cl. 5 to cl. 0 (Fig. 2e) indicating a possible transition from Ly6D-expressing pDC-biased precursors to pre-cDCs.

**CD11c+Siglec-H+Zbtb46+Ly6D+ BM cells have an intermediary phenotype connecting pDC precursors with pre-cDCs.**

**Fig. 2 RNA velocity analysis links Ly6D⁺Siglec-H⁺ precursors and pre-cDCs.** Single-cell RNA sequencing was performed for CD115⁺ CDP, CLP, Ly6D⁺ Siglec-H⁻ LP, Ly6D⁺Siglec-H⁺ LP, Siglec-H⁻ pre-cDC, lo-lo, lo-hi precursors and pDCs sorted from Lineage-depleted BM cells. **a** Diffusion map of 675 DC precursors and related cells with cells highlighted by their identity according to cell sorting. **b** *Ly6d* gene expression overlayed onto the diffusion map. **c** Comparison of *Zbtb46*-eGFP expression in pre-cDC, lo-lo, lo-hi precursors, and pDCs in BM cells of heterozygous *Zbtb46^gfp* mice measured by flow cytometry. Histograms show the *Zbtb46*-eGFP fluorescence signal (normalized to mode), WT cells were used as a negative control. Representative results of 3 independent experiments. **d** RNA velocities projected onto the diffusion map as streamlines with scvelo. Louvain clusters are indicated by colors and numbers. Clusters were annotated according to their gene expression and sorted cell type composition. **e** Partition-based graph abstraction (PAGA) with velocity directed edges computed with scvelo. Solid black arrows indicate probable velocity-inferred transitions of high confidence. Dotted lines indicate clusters that are connected by transcriptome similarity, but do not have sufficient support by RNA velocity to indicate confident transitions. **f** Expression heatmap of manually selected marker genes, scaled between 0 and 1 in Louvain clusters 0 to 6. **g** From left to right: Spliced/unspliced phase portraits, RNA velocity, and expression level of the indicated genes overlaid onto the diffusion map.

To investigate the connection between pDC precursors and pre-cDCs indicated by the scRNA-seq analysis in the context of the whole DC progenitor compartment as well as differentiated DCs, we analyzed the expression of surface markers and transcription factor Zbtb46 in BM cells and splenocytes of heterozygous *Zbtb46^gfp* knockin mice[51] by high dimensional spectral flow cytometry. Zbtb46 expression was used as a marker of cDC committed cells[51]. We found Zbtb46 coexpressed with Ly6D in a fraction of cells which formed a continuum with Ly6D^hiZbtb46⁻ cells in the CD11c⁺Siglec-H⁺ BM compartment (Fig. 3a). When excluding these cells, Ly6D^hiZbtb46⁻ cells still contained lo-lo and lo-hi precursors although at a lower percentage than with the prior gating strategy. The lo-lo precursors, lo-hi precursors, and pDCs were pre-gated as Ly6D^hiZbtb46⁻ cells to exclude any Zbtb46⁺ pre-cDC (Fig. 3a). This refined gating strategy was used from now on.

The concatenated data of BM cells and splenocytes was visualized by UMAP, DC subsets and their precursors were gated and extracted and UMAP was rerun on the extracted data (Fig. 3b, c). Manually gated populations were mapped onto the UMAPs (Fig. 3d and Supplementary Fig. 6, for gating see Supplementary Fig. 5). In the BM, a continuum of cells was observed from Ly6D⁺Siglec-H⁻ LP (population no. 7) to Ly6D⁺ Siglec-H⁺ LP (no. 6) as defined by Rodrigues et al.[40] containing also the pre-pDC (no. 2) described by Dress et al.[39] to lo-lo (no. 4) and lo-hi precursors (no. 3) followed by CCR9^hi BM pDC (no. 1) (Fig. 3d). These continuously distributed cells showed a gradual upregulation of Siglec-H, CD11c, B220, and CCR9 consistent with the major differentiation path to pDCs (Fig. 3e, f). The Zbtb46⁺Ly6D⁺ intermediate cells within the CD11c⁺ Siglec-H⁺ BM fraction (no. 5), which also expressed CX3CR1 (Fig. 3e, f), mapped in between the lo-lo/lo-hi precursors (no. 4, 3) and the Siglec-H⁺(Zbtb46⁺Ly6D⁻) pre-cDC (no. 8) (Fig. 3d). These exhibited a continuous distribution to Siglec-H⁻ pre-cDC (no. 9) followed by BM pre-cDC2 (no. 10) and cDC2 (no. 11). Siglec-H⁻ pre-cDC were also connected to pre-cDC1 (no. 12) and cDC1 (no. 13) in the BM. In the combined UMAP BM cDC1 (no. 13) and BM cDC2 (no. 11) were found adjacent to splenic cDC1 (no. 18) and cDC2 (no. 17) (Fig. 3d), which expressed high levels of Zbtb46, CD11c and MHC class II (Fig. 3e, f). BM pDCs (no. 1) were positioned close to splenic pDCs (no. 16) as expected.

CD11c⁺ cells with a phenotype ranging from pDC-like to cDC2-like have been described as transitional DCs (tDC) found in secondary lymphatic organs but not in the BM[45]. These tDCs are defined as CX3CR1⁺CD11b⁻XCR1⁻ expressing low to high levels of CD11c and Siglec-H[45]. In the UMAP CD11c^low tDC were positioned between pDCs and pre-cDC2, while CD11c^high tDC projected closer to cDC2 (Fig. 3d and Supplementary Fig. 6b). This is consistent with the data shown by Leylek et al.[45]. Using the same gating strategy as for splenic tDCs (shown in Supplementary Fig. 5h), we found cells with a phenotype resembling CD11c^low tDCs also in the BM. Cells gated as

Zbtb46⁺Ly6D⁺ intermediary cells in the BM (shown in Fig. 3a and Supplementary Fig. 5g) partially overlapped with these tDC-like cells (Supplementary Fig. 6a). These results indicate that cells with an intermediary pDC/cDC phenotype are continuously distributed between lo-lo/lo-hi precursors and pre-cDCs in the BM and between pDCs and pre-cDCs (mainly pre-cDC2) in the spleen.

**The Ly6D^hiZbtb46⁻ lo-lo precursor fraction retains the ability to generate cDCs via a Zbtb46⁺Ly6D⁺ intermediary stage.** Our results from scRNA velocity analysis and flow cytometric phenotyping suggested that Ly6D^hiZbtb46⁻ lo-lo precursors can differentiate into cDCs via a Zbtb46⁺Ly6D⁺ intermediary population. We, therefore, investigated the cell fate of Zbtb46⁺Ly6D⁺ cells, lo-lo and lo-hi precursors (sorted as Ly6D^hiZbtb46⁻ cells to exclude any pre-cDCs) as well as Siglec-H⁺, Siglec-H⁻ pre-cDCs and pDCs (sorted from BM cells as shown in Fig. 3a) after 3 days of culture with Flt3L. The frequencies of pDCs defined as CD11c⁺Siglec-H⁺ Ly6D^hiZbtb46⁻CCR9^hi B220^hi and cDCs defined as CD11c⁺ Siglec-H⁻Ly6D⁻Zbtb46⁺MHCII⁺ were determined in the progeny (Fig. 4a). Zbtb46⁺Ly6D⁺ cells downregulated Ly6D and Siglec-H and gave rise to cells with a cDC phenotype comparable to that of cDCs generated from Siglec-H⁻ and Siglec-H⁺ pre-cDCs (Fig. 4b). Lo-lo precursors, sorted as Ly6D^hiZbtb46⁻, generated pDCs and cDCs. Lo-hi precursors predominantly gave rise to pDCs and only few cDCs (Fig. 4a). UMAP analysis confirmed the phenotypic overlap of cDCs generated from the different precursors after 3 days of culture (Fig. 4c). The progeny of both lo-lo and lo-hi precursors produced IFN-α in response to CpG-A stimulation consistent with the generation of functional IFN-producing pDCs (Supplementary Fig. 7a). Thus, CD11c⁺Siglec-H⁺ lo-lo cells rigorously sorted as Ly6D^hiZbtb46⁻ can still generate cDCs in addition to pDCs indicating that high level expression of Ly6D in Siglec-H⁺ precursor cells does not exclude cDC cell fate.

Cells with an intermediary Zbtb46⁺Ly6D⁺ phenotype were also generated from lo-lo and to a small extent from lo-hi precursors (Fig. 4b), indicating that a part of these precursors differentiated into cDCs via a Zbtb46⁺Ly6D⁺ transitional state. To determine the kinetics of this transition we analyzed the expression of pDC and cDC transcriptional regulators and signature genes in precursor cultures with Flt3L after 1, 2, and 3 days compared to that of freshly isolated differentiated BM pDCs and cDC (Fig. 4d). TFs involved in cDC differentiation *Spi1, Irf4, Id2* as well as *Itgax* and *Cd74* were rapidly upregulated in the progeny of Zbtb46⁺ Ly6D⁺ cells and lo-lo precursors. At the same time TFs involved in pDC differentiation *Irf8, Spib, Zeb2, Tcf4* as well as *Ly6d* and *Siglech* were downregulated in both populations with a slightly faster kinetic in the progeny of Zbtb46⁺Ly6D⁺cells. These results are consistent with the generation of cDCs from a part of the lo-lo precursors via Zbtb46⁺Ly6D⁺ intermediate cells.

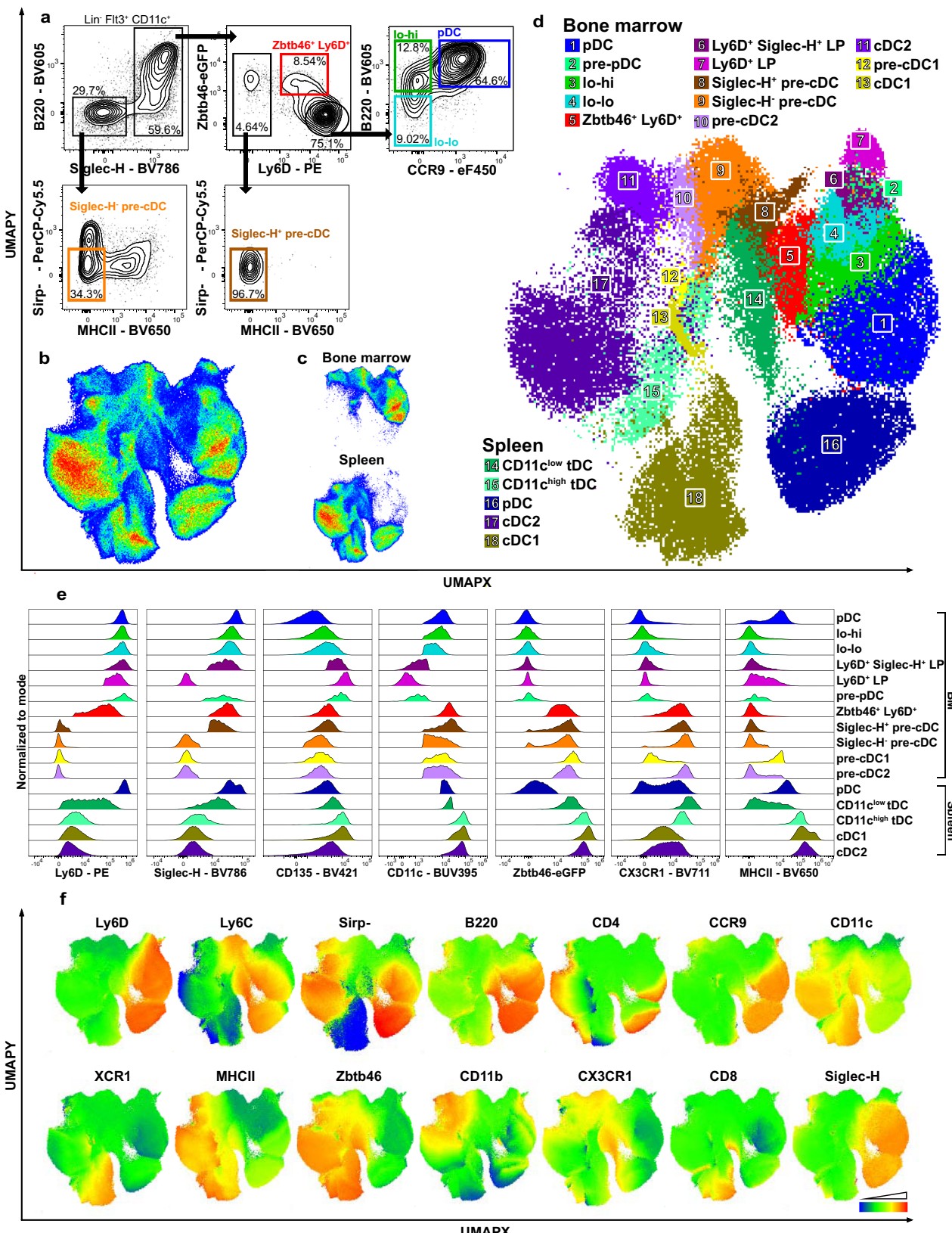

This raised the question, whether Ly6D^{hi}Zbtb46^− lo-lo cells have dual pDC/cDC potential or contain pDC- and cDC-primed precursors. We noticed that cell division had occurred in the progeny of pre-cDC subsets, Zbtb46^+Ly6D^+ cells and a part of the lo-lo cells, but not in cells derived from lo-hi precursors and pDCs after 3 days of culture (Fig. 4e). We analyzed the expression

of Zbtb46 and surface markers in undivided cells after 3 days of culture. Upregulation of Zbtb46 and MHC II, as well as reduced expression of Ly6D and Siglec-H in the undivided fraction, indicated that a part of the Ly6D^{hi}Zbtb46^− lo-lo precursors had started to differentiate into cDCs before the first division and subsequently expanded (Fig. 4f). Indeed, upregulation of *Zbtb46-*

**Fig. 3 CD11c⁺Siglec-H⁺Zbtb46⁺Ly6D⁺ BM cells have an intermediary phenotype connecting pDC precursors with pre-cDCs. a** Siglec-H⁻ pre-cDC, Siglec-H⁺ pre-cDC, Siglec-H⁺Zbtb46⁺Ly6D⁺ cells and lo-lo, lo-hi, and pDC (gated Siglec-H⁺Zbtb46⁻Ly6D⁺) within Lin⁻Flt3⁺CD11c⁺ cells were identified in Lineage-depleted BM cells of heterozygous Zbtb46^gfp mice. **a–f** Lineage-depleted BM and spleen cells of heterozygous Zbtb46^gfp mice were analyzed by multidimensional spectral flow cytometry (25 parameters). **b** The data of BM and spleen cells (1.5 × 10⁶ cells each) were concatenated and subjected to UMAP analysis. The subset containing DC and precursor populations was gated from a parent UMAP that was generated after exclusion of T cells, B cells, NK cells, macrophages, and myeloid progenitors. UMAP was rerun on the extracted data. **c** Distribution of BM and spleen cells in the UMAP of (**b**). **d** Manually gated BM and spleen DC subsets, precursors and Zbtb46⁺ Ly6D⁺ cells indicated by colors and numbers were projected onto the UMAP shown in (**b**). **e** Histograms showing cell surface marker expression for all populations of interest in BM (top) and spleen (bottom), normalized to mode. **f** Log₂ normalized surface marker expression projected onto the UMAP using a color scale from blue (low expression) to red (high expression). Representative results of one of three independent experiments are shown.

eGFP and downregulation of Ly6D expression were observed in approximately 8% of lo-lo derived cells as early as 20 h after start of the culture and before the first cell division (Fig. 4g, h). Cell fate decision before division argues against dual potential and shows that the Ly6D^hiZbtb46⁻ lo-lo precursor fraction contains cDC-primed cells in addition to pDC precursors (see model in Supplementary Fig. 9k).

To investigate the developmental relationship of lo-lo and lo-hi precursors with previously published Ly6D⁺Siglec-H⁺ LPs[40] and CDPs (CD115⁺IL7R⁻) we cultured these for 4 days with Flt3L and analyzed the phenotype of their progeny. As expected, CDP progeny lacked Ly6D expression and had either cDC or pre-cDC phenotype (Supplementary Fig. 8b). In contrast, the progeny of Ly6D⁺Siglec-H⁺ LPs contained lo-hi precursors, pDCs, cDCs and Zbtb46⁺Ly6D⁺ intermediary cells and few but detectable lo-lo precursors (Supplementary Fig. 8c, d). Ly6D⁺Siglec-H⁺ LPs had upregulated CD11c and acquired lo-lo, lo-hi and pDC phenotypes already after 24 h of culture (Supplementary Fig. 8d). At later timepoints a shift from lo-hi cells to pDCs was observed. Zbtb46⁺Ly6D⁺ cells emerged before an accumulation of cDCs was detected (Supplementary Fig. 8d) confirming that the Zbtb46⁺Ly6D⁺ stage is intermediary. Thus, Ly6D⁺Siglec-H⁺ LPs first gave rise to pDCs via lo-lo and lo-hi cells and later also to cDCs via the Zbtb46⁺Ly6D⁺ intermediary stage.

**Ly6D^hiZbtb46⁻ lo-lo precursors give rise to pDCs, Zbtb46⁺Ly6D⁺ cells, and cDCs in the spleen after adoptive transfer.** To interrogate the cell fate of Ly6D^hiZbtb46⁻ lo-lo and lo-hi precursors in vivo in the steady state, these populations were sorted from CTB-labeled BM cells of heterozygous Zbtb46^gfp mice and injected i.v. into untreated non-irradiated CD45.1 congenic mice (Fig. 5a). After 5 days the progeny of lo-lo precursors in the spleen had acquired the phenotype of pDCs, cDC2 and very few cDC1 as well as cells with Zbtb46⁺Ly6D⁺ and tDC phenotype while lo-hi precursors had differentiated almost exclusively into pDCs (Fig. 5b and Supplementary Fig. 7c). UMAP analysis of the combined data of recipient Lin⁻CD11c⁺ splenocytes and the progeny of transferred lo-lo precursors showed the phenotypic similarity of the donor-derived cells with pDCs, Zbtb46⁺Ly6D⁺ cells, tDCs, cDC2 and cDC1 in the spleen (Fig. 5c, d). Notably, donor-derived cDCs, but not pDCs, showed evidence of CTB dilution indicating cell proliferation (Fig. 5e). For comparison, Zbtb46⁺Ly6D⁺ intermediary cells isolated from the CD11c⁺Siglec-H⁺ BM fraction were also transferred and the phenotype of their progeny was analyzed in the spleen after 3 days. Approximately 60% of donor-derived cells in spleen had acquired a cDC phenotype and the remaining cells had maintained the Zbtb46⁺Ly6D⁺ phenotype (Fig. 5f, g). These results confirm the cell fates observed in vitro and support the trajectory from cells with Ly6D^hiZbtb46⁻ lo-lo phenotype to cDCs via a Zbtb46⁺Ly6D⁺ intermediate state.

**Type I IFN limits cDC output and promotes pDC differentiation from precursors.** TLR ligands and type I IFN have been shown to influence pDC and cDC development from BM progenitor cells[54,55]. Experiments performed with lo-lo and lo-hi precursors isolated by cell sorting using the initial gating strategy (shown in Fig. 1a) indicated that type I IFN and TLR9 ligand CpG-A reduced cDC output and increased pDC output from lo-lo and lo-hi precursors. The effect of CpG-A on pDC and cDC output was dependent on IFN-α/-β receptor signaling in the precursors (Supplementary Fig. 9a–d) suggesting an autocrine and/or paracrine effect of type I IFN on cell fate, which could be instructive or selective.

We, therefore, investigated the influence of type I IFN stimulation on the generation of pDCs and cDCs from Ly6D^hiZbtb46⁻ lo-lo and lo-hi precursors subsets in comparison with Zbtb46⁺ Ly6D⁺ cells and pre-cDC subsets in Flt3L cultures. The addition of IFN-α increased the generation of pDCs from Ly6D^hiZbtb46⁻ lo-lo and lo-hi precursors while inhibiting the generation of cDCs within 3 days of culture (Fig. 6a, b and Supplementary Fig. 9e, g). The percentage and an absolute number of Zbtb46⁺Ly6D⁺ cells tended to be increased in cultures containing IFN-α (Fig. 6c, d and Supplementary Fig. 9i). The frequency of divided pDCs was very low and not influenced by type I IFN (Supplementary Fig. 9f), but the frequency of divided cDCs generated from the different precursors was greatly reduced by the addition of IFN-α to the cultures (Supplementary Fig. 9h). Despite inhibition of cell division by IFN-α, upregulation of Zbtb46-eGFP and MHC II and a slightly reduced expression of Ly6D and Siglec-H were still observed in the progeny of IFN-α-treated lo-lo precursors (Fig. 6d, e) consistent with transition to an intermediary Zbtb46⁺Ly6D⁺ phenotype without further expansion. Addition of IFN-α also significantly increased the percentage of CCR9^hiB220^hi pDCs within CD11c⁺Siglec-H⁺Ly6D⁺ cells obtained from lo-lo and lo-hi precursors after 3 days of culture (Fig. 6f, g). Thus, IFN-α promoted pDC versus cDC output from lo-lo and lo-hi precursors by enhancing pDC differentiation and arresting cDC-primed cells in the Zbtb46⁺Ly6D⁺ intermediate state thereby preventing their expansion and full differentiation into cDCs.

**Discussion**

Here we set out to elucidate the heterogeneity and commitment of subsets within the Ly6D⁺Siglec-H⁺ DC precursor fraction in mouse BM. Our data indicate that final differentiation of pDCs from Ly6D⁺Siglec-H⁺ LP occurs in the BM via Ly6D⁺Siglec-H⁺CD11c⁺CCR9^loB220^lo precursors that upregulate B220, which marks the immediate pDC precursor stage. However, the ability to generate cDCs was retained in Ly6D⁺Siglec-H⁺ LP and CD11c⁺Siglec-H⁺ Ly6D^hiZbtb46⁻ lo-lo cells. We show here that the Ly6D⁺Siglec-H⁺ precursor pool is heterogeneous and contributes to both pDC and cDC generation which occurs via a Zbtb46⁺Ly6D⁺ intermediate state and that pDC versus cDC output from these precursors is regulated by type I IFN.

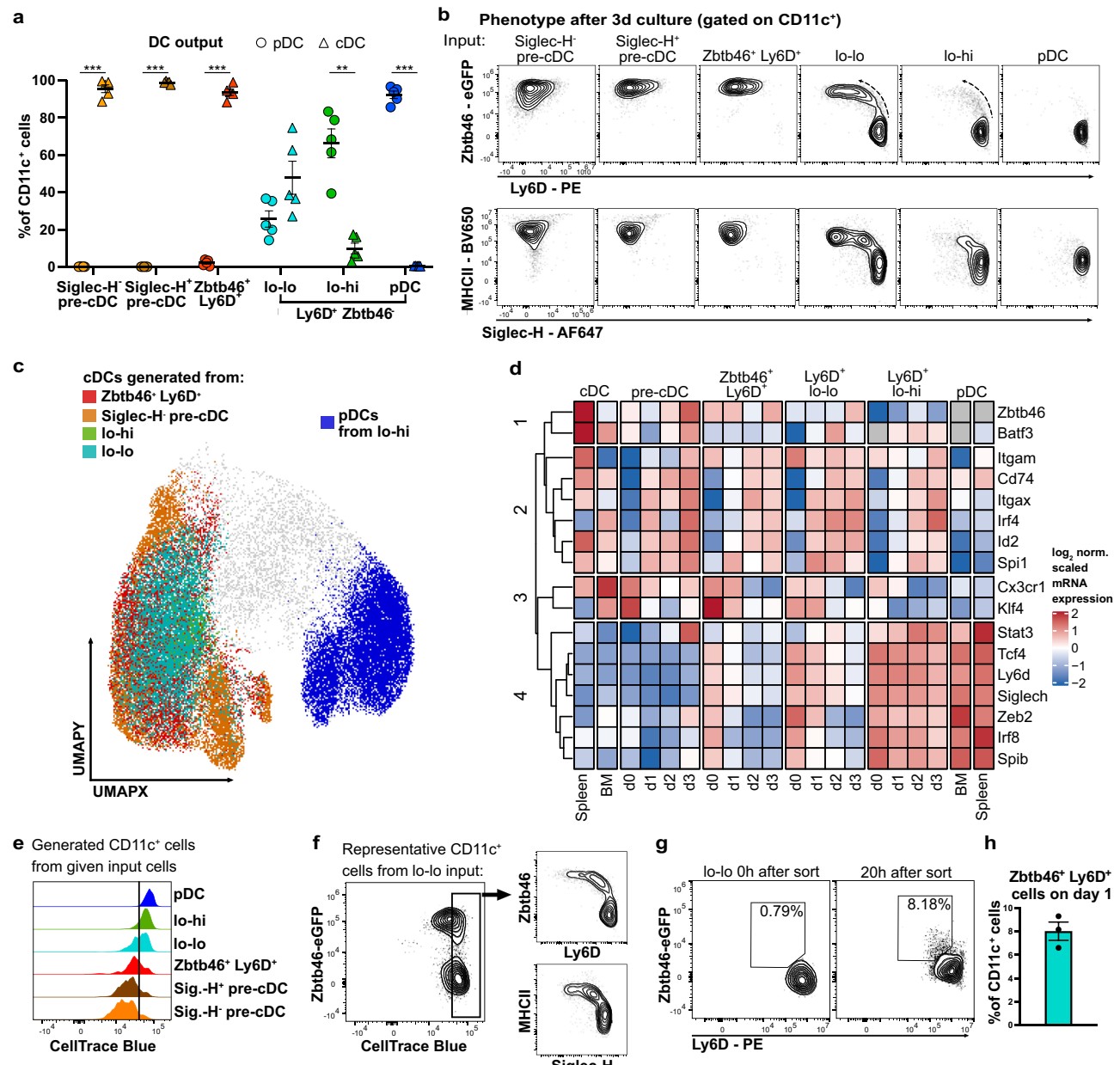

**Fig. 4 CD11c⁺Siglec-H⁺Ly6DʰⁱZbtb46⁻CCR9ˡᵒB220ˡᵒ precursors contribute to cDCs via a Zbtb46⁺Ly6D⁺ intermediary stage. a** Siglec-H⁻ pre-cDC, Siglec-H⁺ pre-cDC, Siglec-H⁺Zbtb46⁺Ly6D⁺ cells as well as lo-lo, lo-hi precursors and pDCs (gated Siglec-H⁺Ly6DʰⁱZbtb46⁻ as shown in Fig. 3a) were sorted from Lineage-depleted BM cells of heterozygous *Zbtb46gfp* mice and cultured for 3 days with Flt3L on EL08-1D2 stromal cells. The phenotype of the progeny was analyzed by flow cytometry. Percentages of pDCs (circles) or cDCs (triangles) within CD11c⁺ progeny are shown (mean ± SEM, *n* = 5); pDC vs cDC output was compared using paired, two-sided t-tests with Holm-Šídák correction for multiple testing. Adjusted *p*-values: <0.05(*), <0.005(**), <0.0.001(***). **b** Phenotype of cells derived from the indicated populations after 3 days of culture (representative results, *n* = 5). A potential transition from Ly6D⁺Zbtb46⁻ to Zbtb46⁺Ly6D⁻ is indicated by the dotted arrows. **c** The indicated populations were sorted and cultured for 3 days with Flt3L without stromal cells, then analyzed by flow cytometry. UMAP analysis of concatenated CD11c⁺ cells from all samples. cDCs generated from each precursor subset were gated as CD11c⁺Zbtb46ʰⁱᵍʰMHCIIʰⁱᵍʰ and projected onto the UMAP, together with pDCs generated from lo-hi precursors for comparison. **d** Heatmap of hierarchically clustered log₂ normalized relative gene expression (scaled per gene) in the progeny of the indicated DC precursors before or after 1, 2 and 3 days of culture with Flt3L measured by qRT-PCR (grey: not detectable). Gene expression in freshly isolated BM and spleen pDC and cDC served as a reference (mean values of 2 independent experiments are shown). **e** CellTrace Blue proliferation dye signal in the CD11c⁺ fraction of the indicated input cells after 3d of culture (representative results, *n* = 5). **f** Expression of the indicated markers in the undivided fraction of CD11c⁺ cells generated from lo-lo precursors after 3 days. **g** Zbtb46 and Ly6D expression in lo-lo precursor cells directly after sorting and after 20 h of culture with Flt3L. The precentage of Zbtb46⁺Ly6D⁺ cells is shown in the gates. **h** Percentage of Zbtb46⁺Ly6D⁺ cells within CD11c⁺ cells derived from lo-lo precursors after 20 h of culture (mean ± SEM, *n* = 3).

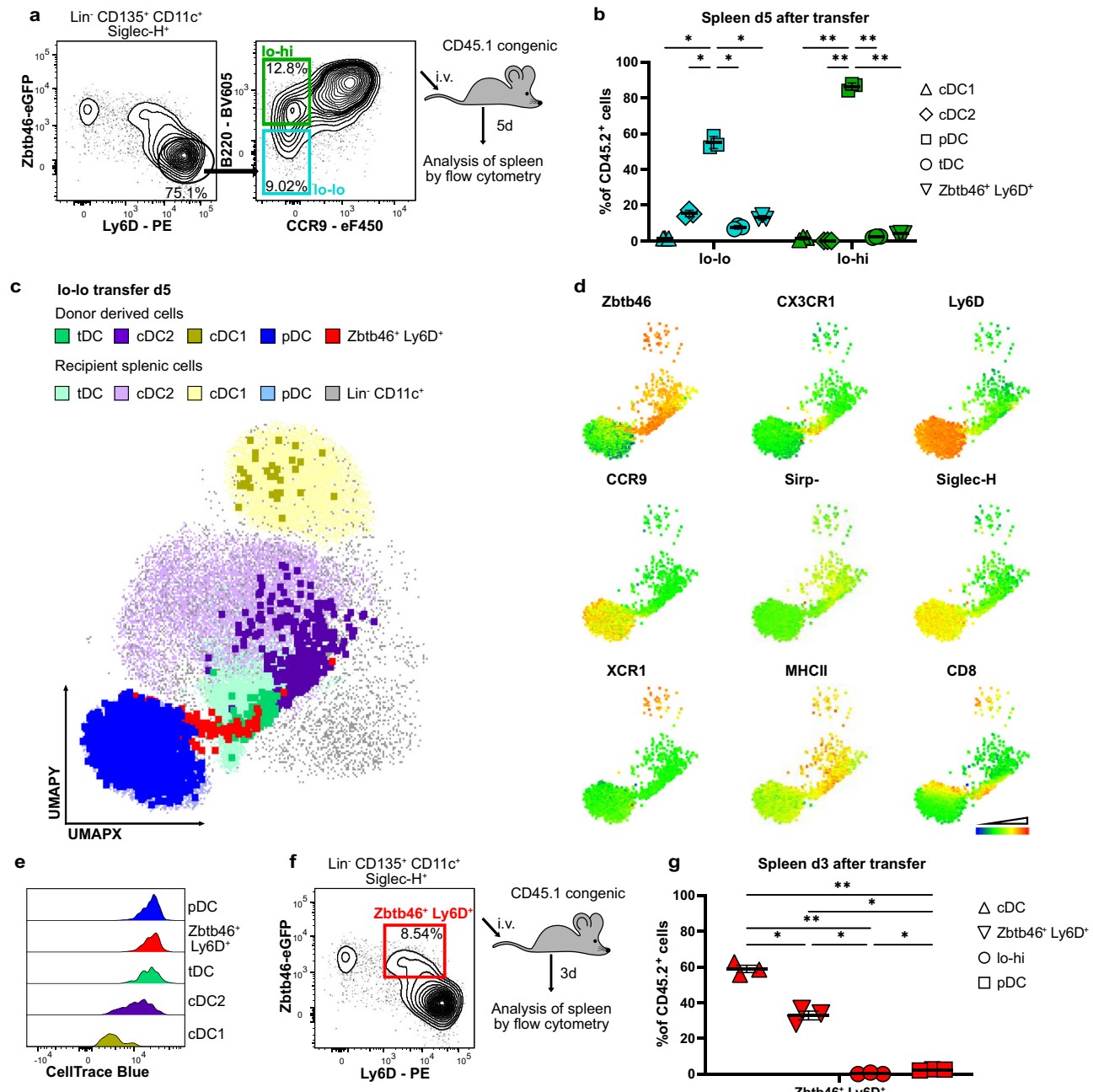

**Fig. 5 Cell fate of Siglec-H⁺Ly6DʰⁱZbtb46⁻CCR9ˡᵒB220ˡᵒ and CCR9ˡᵒB220ʰⁱ precursors after transfer in vivo. a** Siglec-H⁺Ly6DʰⁱZbtb46⁻ lo-lo and lo-hi cells were sorted from BM of heterozygous *Zbtb46ᵍᶠᵖ* mice and transferred i.v. into congenic CD45.1 mice. After 5 days splenocytes were analyzed by multiparameter spectral flow cytometry. **b** Phenotype of recovered donor derived cells on day 5 after injection depicted as percentage of total CD45.2⁺ recovered cells (*n* = 3, mean ± SEM). **c** From each recipient of lo-lo precursors 400,000 Lin⁻ CD11c⁺ recipient cells and all donor-derived cells were concatenated and UMAP dimensionality reduction was performed. DC subsets were manually gated for both recipient and donor-derived cells and projected onto the UMAP. Donor-derived cells are highlighted as large dots. **d** Log₂ normalized surface marker and Zbtb46 expression in donor-derived cells overlaid onto the UMAP of (**c**) using a color scale from blue (low expression) to red (high expression). **e** Histograms showing fluorescence intensity of CellTrace Blue dye (normalized to mode) in cells derived from transferred lo-lo precursors after 5 days. **f** Siglec-H⁺Zbtb46⁺Ly6D⁺ cells were sorted from BM of heterozygous *Zbtb46ᵍᶠᵖ* mice and transferred i.v. into congenic CD45.1 mice. After 3 days splenocytes were analyzed by multiparameter spectral flow cytometry. **g** Phenotype of recovered donor derived cells on day 3 after injection depicted as percentage of total recovered cells (*n* = 3, mean ± SEM). **b**, **g** Output phenotypes were compared using a two-way ANOVA with Šídák's correction for multiple testing. Adjusted *p*-values: <0.05(*), <0.005(**).

The results are consistent with a model of Ly6Dʰⁱ Siglec-Hᵗ Zbtb46⁻ lo-lo precursors containing pDC-primed cells, which rapidly transition to pDCs by upregulating B220 and CCR9 without dividing, as well as cDC-primed cells that acquire a Zbtb46⁺Ly6D⁺ phenotype and then fully differentiate into cDCs while dividing. In the presence of IFN-α stimulation, cells

engaged in cDC generation (indicated by upregulation of Zbtb46) failed to divide and fully differentiate into cDCs (model B, shown in Supplementary Fig. 9k). The generation of pDCs without cell division and the early emergence of Zbtb46 expression before cell division argues against dual pDC/cDC potential in individual lo-lo precursors (model A, Supplementary Fig. 9k). We can exclude

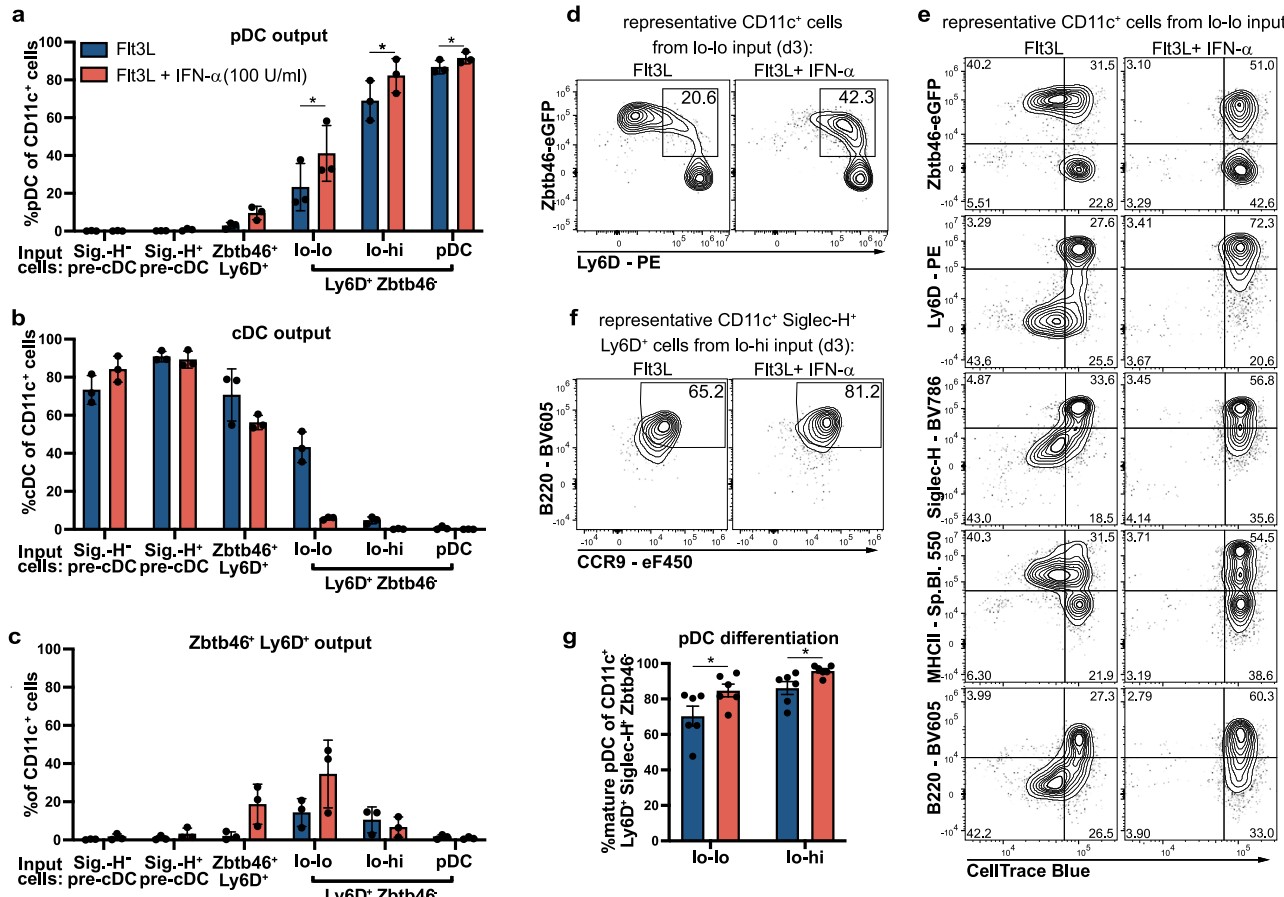

**Fig. 6 Modulation of pDC versus cDC output from CD11c⁺Siglec-H⁻Ly6D⁺Zbtb46⁻CCR9ˡᵒB220ˡᵒ precursors by type I IFN in vitro.** Siglec-H⁻ pre-cDC, Siglec-H⁺ pre-cDC, Siglec-H⁺Zbtb46⁺Ly6D⁺ cells and lo-lo, lo-hi and pDC (gated Siglec-H⁺Ly6D⁺Zbtb46⁻) were sorted from Lineage-depleted BM cells of heterozygous *Zbtb46*ᵍᶠᵖ mice and cultured for 3 days with Flt3L only or Flt3L and 100 U/ml IFN-α on EL08-1D2 stromal cells. Percentages of pDCs **a**, cDCs (**b**), and Zbtb46⁺Ly6D⁺ cells (**c**) within CD11c⁺ progeny after 3 days are shown for each input population as mean ± SEM (*n* = 3). **d** Zbtb46 vs. Ly6D expression in CD11c⁺ cells derived from lo-lo precursors with Flt3L alone (left) and Flt3L/IFN-α (right); representative results of three experiments. **e** Expression of Zbtb46 and several surface markers vs. CTB signal. Representative example of CD11c⁺ cells generated from lo-lo precursors after 3 days of culture with Flt3L alone (left) and Flt3L/IFN-α (right, *n* = 3). **f** Representative example of B220 vs. CCR9 expression and pDC gating of CD11c⁺Siglec-H⁺Ly6D⁺ cells derived from lo-hi precursors after 3d of culture (*n* = 5). **g** Percentage of differentiated pDCs within CD11c⁺Siglec-H⁺Ly6D⁺Zbtb46⁻ cells derived from lo-lo and lo-hi precursors after 3d of culture with and without IFN-α addition (mean ± SEM, *n* = 5). Conditions with and without IFN-α were compared for each input population using paired, two-sided t-tests with Holm–Šídák correction for multiple testing. Adjusted *p*-values: <0.05(*).

that contamination with Zbtb46⁺ pre-cDCs led to the generation of cDCs in the culture of lo-lo cells because lo-lo cells were sorted as Ly6DʰⁱZbtb46⁻ cells and showed de novo expression of Zbtb46 in undivided cells. The pDC- and cDC-primed cells within the Ly6DʰⁱSiglec-H⁺Zbtb46⁻ lo-lo precursor fraction may originate from an earlier common clonal progenitor or from separate progenitors which have a similar phenotype and gene expression profile at this stage.

Our targeted scRNA-seq approach allowed us to position CD11c⁺Siglec-H⁺Ly6Dʰⁱ lo-lo and lo-hi precursors on the pDC trajectory between Ly6D⁺Siglec-H⁺ LP[39,40] and differentiated pDCs using methods of trajectory inference based on similarity of gene expression[41] and transcriptional dynamics[52]. RNA velocity analysis of the data set identified progenitor cells with either B cell or pDC cell fate within the IL7-R⁺Ly6D⁺Siglec-H⁻ fraction confirming previous reports[39–41]. Hermans et al. identified cells with either B cell or pDC bias within CLPs with the latter expressing higher levels of Irf8 and Cd34[41]. Rodrigues et al. showed that the fraction of Ly6D⁺Siglec-H⁻ LPs expressing high levels of Irf8 and low levels of Ebf1 generated almost exclusively pDCs but not B cells in culture with Flt3L[40]. We confirmed

differential expression of Irf8 and Ebf1 within this LP fraction in our data set. In addition, RNA velocity analysis indicated upregulation of Flt3 in Ly6D⁺Siglec-H⁺ progenitors, but downregulation of Flt3 in Ly6D⁺Siglec-H⁻ progenitors which concomitantly upregulated transcription factors involved in B cell development including Ebf1 and Pax5. Repression of Flt3 by Pax5 was shown to be essential for B cell generation beyond the pre-pro B cell stage[53].

Results of our high-dimensional single-cell gene expression and surface marker expression analyses showed a link between Ly6D⁺Siglec-H⁺ and pre-cDCs, which was formed by CD11c⁺Siglec-H⁺ cells expressing both Ly6D and Zbtb46. Differentiation assays and adoptive transfer experiments demonstrated the ability of CD11c⁺Siglec-H⁺Ly6DʰⁱZbtb46⁻ lo-lo precursors to generate Zbtb46⁺Ly6D⁺ intermediate cells and subsequently Zbtb46⁺Ly6D⁻ cells with cDC phenotype. BM CD11c⁺Siglec-H⁺Zbtb46⁺Ly6D⁺ cells showed cDC potential in vitro and in vivo, thus confirming this developmental trajectory. Zbtb46 is a transcriptional repressor that maintains cDC quiescence in the steady-state and preserves the cDC2/cDC1 ratio in lymphoid organs but is not required for cDC development[22,51,56]. It is therefore unlikely that Zbtb46-

haploinsufficiency influenced the generation of cDCs from precursors expressing Zbtb46 in our assays.

Ly6D[+]Siglec-H[+] LP[40] differ from the CD11c[+]Siglec-H[+]Ly6D[hi]Zbtb46[−] lo-lo and lo-hi precursors by lack of CD11c expression and a higher proliferative potential. Ly6D[+]Siglec-H[+] LP cultured with Flt3L rapidly upregulated CD11c and generated cells with lo-lo, lo-hi and pDC phenotype as well as Zbtb46[+]Ly6D[+] intermediate cells and cDCs. These results confirmed the proposed hierarchy that Ly6D[+]Siglec-H[+] LP are upstream of the lo-lo and lo-hi precursors and contribute to cDCs via a Zbtb46[+]Ly6D[+] intermediate stage. Interestingly, Rodrigues et al. had also observed cDC output from Ly6D[+]Siglec-H[+] LP in the range of 10–20%[40]. The higher output of cDCs from this population in our experiments could be due to differences in culture conditions which may influence the dynamics and outcome of the differentiation assays. The failure of CD115[+]IL-7R[−] CDPs to generate lo-lo or lo-hi precursors or Zbtb46[+]Ly6D[+] cells further confirmed that these cells originate from Ly6D[+]Siglec-H[+] LPs and not from CDPs as inferred from the single-cell RNA velocity analysis.

CD11c[+]Siglec-H[+]Ly6D[+] pDC-like cells expressing Zbtb46 and CX3CR1 were described in BM and spleen, which share functional properties of pDCs (IFN-α production) and cDCs (antigen presentation) and were considered to be a subset of mature pDCs[40]. These were shown to be transcriptionally similar to BST2[+]Siglec-H[+] cells derived from CD115[+] CDPs in vitro suggesting a CDP origin of these Zbtb46[+] pDC-like cells[40]. We found no evidence for a CDP-origin of CD11c[+]Siglec-H[+]Ly6D[+]Zbtb46[+] cells and did not detect pDC generation from CD115[+]IL-7R[−] CDPs, which lack Ly6D, in agreement with Dress et al.[39].

We found cells resembling the CD11c[low] tDCs described by Leylek et al.[45] in spleen also in the BM. These partially overlapped with the cDC-committed Zbtb46[+]Ly6D[+] intermediary cells studied here. Adoptively transferred lo-lo precursors generated progeny with continuous distribution of phenotypes between pDCs and cDCs which encompassed cells with Zbtb46[+] Ly6D[+] and tDC phenotype. This raises the possibility that progeny of cDC-committed lo-lo precursors transit through Zbtb46[+]Ly6D[+] and tDC stages on their way to cDCs. It was shown that cells with a similar transitional phenotype isolated from mouse BM and spleen produce cytokines upon stimulation and present antigens to T cells[40,45]. Similarly, human tDC (also designated Axl[+] DC or pre-DC) isolated from blood were shown to be functional APCs[45,47] but can still transit to cDCs[46]. Even though our in vitro and in vivo data indicate that CD11c[+]Siglec-H[+]Zbtb46[+] Ly6D[+] cells further differentiate into cDCs in the steady state they may actively participate in immune responses even before full differentiation.

We did not observe generation of CD11c[+]Siglec-H[+]Zbtb46[+]Ly6D[+] cells or cDCs from differentiated CCR9[hi] pDCs, indicating stability of the pDC phenotype at this stage under steady state conditions. However, it was reported that human bona fide pDCs converted to tDCs (Axl[+] DC) and subsequently cDC-like cells upon stimulation with CD40L[57]. Similarly, during MCMV infection in mice, pDCs progressively acquired the transcriptome profile of tDC/cDCs and the capacity to present antigens to T cells[58]. Thus, plasticity is maintained even in fully differentiated pDCs in humans and mice under stimulatory conditions.

Type I IFN has been shown to enhance pDC generation from mouse CLPs[54,37]. Here we show that type I IFN also influenced pDC versus cDC differentiation at a later stage in Ly6D[+]Siglec-H[+]CD11c[+]CCR9[lo] precursors, where type I IFN can be produced[43] and is sensed by the precursors themselves. The addition of IFN-α to Flt3L cultures limited cDC output and promoted pDC output

from lo-lo precursors. This was mainly a selective effect, because cDC-primed cells within this precursor fraction still upregulated Zbtb46 and started to downregulate Ly6D, but did not further proliferate and differentiate into cDCs thereby reducing the number of cDCs generated.

At the same time, we observed an enhanced differentiation into CCR9[hi]B220[hi] pDCs in the presence of IFN-α. In line with this observation from in vitro cultures, we detected higher expression of genes involved in the induction of, and response to, type I IFN in BM pDCs compared to lo-hi precursors, indicating an involvement of type I IFN signals during this final differentiation step. Consistently, type I IFN was also shown to promote functional maturation of human pDCs generated from CD34[+] cells in vitro[59], and self-priming by constitutively produced IFN I was demonstrated to be important for full functionality of human pDCs[60].

We show that pDCs are generated from Ly6D[+]Siglec-H[+] LPs via advanced precursors expressing CD11c, Siglec-H and subsequently B220. However, high level expression of Ly6D and Siglec-H in the DC precursor compartment does not exclude cDC cell fate. Instead, we found that Ly6D[+]Siglec-H[+] LP and CD11c[+]Siglec-H[+]Ly6D[hi]Zbtb46[−] precursors contain cDC-primed cells which generate cDCs under steady state conditions via a Zbtb46[+]Ly6D[+] intermediate stage. Modulation of pDC versus cDC output by factors such as type I IFN suggests that the composition of the DC compartment can still be adapted at later differentiation stages to specific requirements during dynamic immune responses.

## Methods

**Mice.** Wildtype (C57BL/6, originally from Envigo) *Ifnar1*[−/−], *Myd88*[−/−] on C57BL/6 background, and *Ptprc[a]* (CD45.1) congenic mice were originally purchased from The Jackson Laboratory and bred in the Core Facility Animal Models at the Biomedical Center, LMU Munich under pathogen free conditions. Homozygous *Zbtb46[gfp]* knockin mice backcrossed to C57BL/6 for 10 generations were obtained from K. Murphy, Washington University St. Louis and rederived into our SPF facility and mated with C57BL/6 mice to obtain *Zbtb46[gfp]* heterozygous mice[22,51]. Adult mice of both sexes (2–8 months old) were used for isolation of cells for ex vivo analysis and for in vitro and in vivo experiments. All experimental procedures involving mice were performed in accordance with the regulations of, and approved by, the local government (Regierung von Oberbayern, Az. ROB-55.2Vet-2532.Vet_02-17-22).

**Cell lines.** The EL08-1D2 stromal cell line was obtained from R. Oostendorp, University Hospital Rechts der Isar, Technical University Munich[61] and cultured with EL08 medium (MEM-α-Glutamax (Invitrogen, Karlsruhe, Germany), 15% v/v heat inactivated FCS, 5% v/v horse serum (Stem Cell Technologies, Köln, Germany), 1% penicillin/streptomycin, 0.01 mM β-mercaptoethanol).

**Preparation of BM and spleen cells.** BM cells were isolated from femora, tibiae, and hip bones by flushing with RPMI 1640 and passed through a 100 μm strainer. Spleens were cut into pieces and digested for 30 min at 37 °C in DC medium containing DNAse I (Sigma) and Collagenase D (Sigma), then passed through 100 and 40 μm cell strainers. After red blood cell lysis using Red Blood Cell Lysing Buffer (Sigma) for 5 min at RT and washing with RPMI 1640 BM and spleen cells were resuspended in DC medium (RPMI 1640, 10% v/v fetal calf serum (FCS), 1% NEAA (10 mM), 1% Glutamax, 1% sodium pyruvate, 1% penicillin/streptomycin (10.000 U/ml/10.000 μg/ml), 50 μM β-mercaptoethanol) at the required cell density. For sorting of precursor populations BM cells from 2 to 10 sex- and age-matched mice were pooled.

**Isolation of precursor populations by fluorescence-activated cell sorting.** To enrich for populations of interest, Lineage-positive cells were depleted from the BM cell suspension by magnetic cell sorting using APC-Cyanine7 antibodies against CD3, CD19, Ly6G, NK1.1, Ter119 (all antibodies used in this study incl. information on dilution, clone and manufacturer can be found in Supplementary Table 2) and anti-Cy7 beads (Miltenyi Biotech) according to the manufacturer's protocol. For experiments involving Ly6D[+]Siglec-H[+] LP[40], the lineage markers consisted of CD3, CD19, Ly6G, NK1.1, CD105 and CD11c. Where applicable, lineage-depleted BM cells were preincubated with CellTrace Blue proliferation dye (CTB, ThermoFisher) according to the manufacturer's protocols, followed by mouse FcR Blocking Reagent (Miltenyi Biotec) and incubation with viability dyes

and antibodies against cell surface markers diluted in FACS buffer (PBS, 2% FCS, 2 mM EDTA) for 20 min on ice in the dark. After washing with FACS buffer cells were resuspended in FACS buffer and sorted on a BD FACSAria Fusion or a BD FACSAria III cell sorter into tubes containing DC medium or as single cells into 96-well plates containing 5 μl mcSCRBseq lysis buffer (5 M Guanidine HCl, 1% β-ME, 0.2% Phusion HF buffer 5×) per well. Gating strategies for sorting are shown in Supplementary Fig. 3, Fig. 3a and Supplementary Fig. 8.

**Flow cytometry.** Lineage-depleted BM and spleen cell suspensions or cells after in vitro differentiation were stained with viability dyes and antibodies against cell surface markers as described above. Antibodies were purchased from BioLegend, BDBiosciences, eBioscience/Thermo Fisher Scientific (detailed information on antibodies are provided in the Reporting Summary). For high-dimensional flow cytometric analysis of BM and spleen cells from heterozygous *Zbtb46*[gfp] mice, 10 μl of the Lin[+] fraction after magnetic cell sorting was added back to the Lin[−] cell suspensions to allow for a more accurate projection of DC and precursor populations. Samples were measured using either the CytoFLEX S flow cytometer (Beckman Coulter) or the Cytek Aurora Spectral Analyzer. The data were analyzed using FlowJo software v10.7.2 (BD Biosciences). For UMAP analysis, FlowJo plugin UMAP v3.1 was used.

**In vitro differentiation assays.** For cocultures of precursors with Flt3L and stromal cells, EL08-1D2 cells were seeded at a density of $2 \times 10^4$ cells/ml in gelatin-coated 12-well plates in EL08 medium the day before to allow for adherence. The medium was discarded and 500 μl of DC medium was added per well. Precursor cells and pDCs sorted from pooled BM cells of the indicated mouse strains were added at $5 \times 10^4$–$3 \times 10^5$ per well in 250 μl DC medium; 250 μl DC medium containing 30 ng/ml Flt3L (3% supernatant of a Flt3L secreting cell line generated in house from CHO-Flt3L cells, obtained from N. Nicola, WEHI, Australia) and 100 U/ml Universal Type I IFN (PBL Assay Science) or 0.5 μM CpG-A (Eurofins) were added as indicated. The cells were cultured for 3 days at 37 °C and 5% CO$_2$ and harvested for flow cytometric analysis. For gene expression analysis in the progeny the precursor cells were cultured in DC medium with 100 ng/ml Flt3L (10 % Flt3L-supernatant) without stromal cells in 96-well round-bottom plates ($2 \times 10^4$–$8 \times 10^4$ cells per well, 250 μl per well). Sorted BM pDCs and cDCs were used as reference samples. At the indicated time points (d0, d1, d2, and d3) plates were centrifuged, supernatants removed and cells were lysed with RLT Plus lysis buffer from RNeasy Plus Micro Kit (Qiagen, Cat. No: 74034) and frozen at −20 °C. The phenotype of the progeny was also assessed by flow cytometry on day 3.

**In vivo cell transfer.** Zbtb46[+]Ly6D[+] cells, Ly6D[−]Zbtb46[−] lo-lo and lo-hi cells were sorted from BM cells as described above and injected intravenously into tail veins of congenic CD45.1[+] mice. On day 3 or 5 mice were sacrificed and spleens were harvested and digested as described above for flow cytometric analysis.

**Bulk RNA sequencing and data analysis.** RNA isolation was performed using Quick-RNA microprep Kit (Zymo Research, Cat. No: R1050), following the manufacturer's instructions. RNA concentrations were measured using a micro-volume spectrophotometer (SimpliNano, Thermo Fisher Scientific) and adjusted to 5 ng/μl. Libraries were constructed using a slightly modified bulk SCRB-seq[62] protocol, as described before[63]. Sequencing (16+50 base pairs) was performed on the HiSeq1500 platform with a target sequencing depth of 10 million reads per sample. Raw fastq data was processed using the zUMIs pipeline (version 0.0.1[64]). Within zUMIs, barcode and UMI sequences were quality filtered. Remaining reads were mapped to the mouse genome (build mm10) using STAR (version 2.5.2b[65]). Gene identities were obtained from Ensembl annotations (GRCm38 release 84). Count matrices were pre-filtered to eliminate spuriously expressed genes (Mean < 1 count). After initial data processing, data were analyzed using the DESeq2 v.1.18.1[66] package in R (v3.5), following the standard workflow from the package's vignette, with the only variation of using likelihood ratio test (LRT) to test the hypothesis. A cutoff of 0.01 (Benjamini–Hochberg adjusted p-value) was used to select significantly differentially expressed genes (DEGs). For further analyses (PCA, clustering), the data were transformed using a Variance Stabilizing transformation. The 2880 DEGs thus detected were clustered for co-regulation by z-score hierarchical clustering based on Euclidean distances. Clusters were individually analyzed for enrichment of TF binding motifs using the RcisTarget v1.0.2[67] package in R. Clusters were further individually analyzed for enrichment of functional pathways, using the GeneOverlap v1.14.0[68] package in R, and gene sets from the Molecular Signature Database (MSigDB v6.1)[69,70]. A cutoff p-value of 0.05 was used for significance. For gene signature comparisons, genes with expression values greater than the inter-population median were compared with published gene signatures and percentages of congruency were calculated.

**Single-cell RNA sequencing.** The populations of interest were sorted one cell per well across nine 96-well plates, 96 cells per population, with each population spread across two plates (for sort gates see Supplementary Fig. 3). Using shared HVGs for analysis was sufficient to remove batch effects between plates. Libraries for scRNA-seq were prepared following the plate-based mcSCRBseq protocol as described[49]. Single-read 50 bp sequencing was performed on the HiSeq1500 platform with a target sequencing depth of 50,000 reads per cell. Barcode/UMI-filtering, mapping,

and counting of the raw data was performed using the zUMIs pipeline (version 2.5.6[64]). Within zUMIs, barcode sequences were quality filtered, allowing up to 2 bases below Phred quality score of 20. Remaining reads were mapped to the mouse genome (build mm10) using STAR (version 2.6.0a[65]). Gene identities were obtained from Ensembl annotations (GRCm38.75).

**Single-cell RNA-seq data analysis.** Velocity-tagged zUMIs output from the scRNA-seq data was processed using the Python velocyto v0.17[71] pipeline with specified barcodes. The resulting loom file was used for downstream RNA velocity analysis using the scvelo package v0.2.2[52]. Cells with abnormally high or low gene counts were excluded from the analysis. Genes present in less than 10 cells were excluded as well. Cells showing high level expression of mast cell genes (*Prss34*, *Prg2*, and *Mcpt8*) were excluded from the analysis. Cells with a total transcript number below 900 were excluded. Final analysis was performed on 675 cells. The analysis was done on the top 10,000 highly variable genes. The stochastic mode was used for velocity calculation. Cluster identification was done using the Louvain community detection algorithm with an adjusted resolution (1.8) to identify more clusters than with the default value.

**qRT-PCR.** RNA was extracted according to the RNeasy Plus Micro Kit (Qiagen, Cat. No: 74034) protocol. RNA quantity and quality were assessed using a 2100 Bioanalyzer (Agilent Technologies). cDNA was produced from 3 to 10 ng of RNA with the SuperScript III reverse transcriptase (Invitrogen) according to the manufacturer's instructions. cDNA quantity and quality were assessed using a Nano-Drop 1000 Spectrophotometer (ThermoFisher). For each qPCR reaction, 200 ng of cDNA were used with commercially available Taqman probes (see Supplementary Table 1) according to the manufacturer's instructions with HPRT as the housekeeping gene in technical duplicates (LightCycler480, Roche Life Science). The 2^-ddCT method was employed to calculate relative mRNA expression in Microsoft Excel using the mean dCT value for each gene as a reference. Resulting fold-changes were log2 normalized and visualized in a heatmap (scaled per gene) using R package ComplexHeatmap with row_split argument set to 4, resulting in a hierarchical clustering with 4 gene clusters.

**Statistical analysis.** Statistical analysis was done with Microsoft Excel, R and Prism 8 (GraphPad). Paired, two-sided t-test with Holm–Šídák correction for multiple testing was performed unless denoted otherwise. Significant differences were assumed when the adjusted p-value was smaller than 0.05.

**Reporting summary.** Further information on research design is available in the Nature Research Reporting Summary linked to this article.

## Data availability

Bulk RNA-seq data generated in this study have been deposited in the NCBI GEO database with Accession No. GSE189780. ScRNA-seq data that support the findings of this study have been deposited at the European Nucleotide Archive under Accession No. PRJEB52646 and in Array-Express (https://www.ebi.ac.uk/arrayexpress/experiments/E-MTAB-11752/) and in processed form for simplified downstream analysis with scvelo at figshare https://figshare.com/articles/dataset/pDC_precursor_scvelo_h5ad/17013788/1 (https://doi.org/10.6084/m9.figshare.17013788.v1). The mouse reference genome mm10 (GRCm38 release 84) used in this study can be found under GeneBank https://www.ncbi.nlm.nih.gov/assembly/GCF_000001635.20/ under Accession No. GCA_000001635.2. Molecular Signature Database gene sets are available via the respective R package MSigDB. Further data acquired and analyzed in this study are available from the corresponding author upon reasonable request. Source data are provided with this paper.

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

## Acknowledgements

This work is part of the thesis of K.L. This work was funded by the Deutsche Forschungsgemeinschaft (DFG) project-IDs 210592381-SFB 1054-A06 (A.K.), 369799452-SFB/TRR 237-B14 (A.K.), 252441188-KR2199/9-1 (A.K.), 391217598-KR2199/10-1 (A.K.), and 210592381-SFB 1054-B06 (T.K.), 213904703-TRR 128-A07 (T.K.), 408885537-TRR 274-A01 (T.K.), 390857198-EXC 2145 (SyNergy) (T.K.), 278529602-SFB1243-A09 (R.O.). T.K. received additional funding from the European Research Council (ERC GoG 647215, T.K.). E.D. was supported by a Friedrich-Baur foundation grant and E.W. received a scholarship from the Villigst foundation. C.Z., J.B., and W.E. were supported by the Deutsche Forschungsgemeinschaft (DFG) 278529602-SFB 1243-A14 (W.E.). F.J.T. gratefully acknowledges support by the Helmholtz Association's Initiative and Networking Fund through Helmholtz AI, ZT-I-PF-5-01 (F.J.T.) and sparse2big, ZT-I-007 (F.J.T.) and by the Chan Zuckerberg Initiative DAF, advised fund of Silicon Valley Community Foundation, 2019-207271 (F.J.T.). B.U.S. was supported by the European Research Council Starting Grant ERC-2016-STG-71518 (B.U.S.) and the Deutsche Forschungsgemeinschaft (DFG) 360372040–SFB 1335-P08 (B.U.S.) and 322359157-FOR2599-A03 (B.U.S.). We thank Kenneth Murphy for providing *Zbtb46gfp* mice. We acknowledge the Core Facility Flow Cytometry of the Biomedical Center, LMU Munich and thank Lisa Richter and Pardis Khosravani. We acknowledge the Core Facility Animal Models of the Biomedical Center, LMU Munich. We acknowledge the Core Facility Bioinformatics of the Biomedical Center, LMU Munich and thank Tobias Straub for essential support with bioinformatic and statistical analysis. We thank Dominik Alterauge and Dirk Baumjohann for supporting the setup of scRNA-seq protocols and for sharing reagents, and Stefan Krebs and Helmut Blum at Laboratory for Functional Genome Analysis, Gene Center, LMU Munich, for sequencing.

## Author contributions

K.L. and A.M. designed, performed, and analyzed most of the experiments, interpreted data and contributed to writing the manuscript; E.D., C.S., E.W., and L.R. performed experiments and analyzed data; C.Z., J.B., and W.E. provided critical tools and contributed to RNA seq analysis. V.B., M.D.L., and F.J.T. contributed to computational analysis of scRNA-seq data; R.O. provided the EL08 stroma cell line; T.K. designed and supervised some experiments and analyzed data; B.U.S. designed experiments and interpreted data; A.B.K. conceived the project, designed experiments, analyzed and interpreted data and wrote the manuscript. All of the authors read and approved the final version of the manuscript.

## Funding

## Competing interests

F.J.T. reports receiving consulting fees from Roche Diagnostics GmbH, Immunai Inc., Singularity Bio B.V., CytoReason Ltd, and Omniscope Ltd Inc., and has an ownership interest in Cellarity, Inc. and Dermagnostix GmbH. VB is a full-time employee of Cellarity Inc. and reports ownership interest in Cellarity Inc.; the present work was carried out at Helmholtz Munich. The other authors declare no competing interests.
