## [Peer Review File · Nature Communications]

Ly6D+ Siglec-H+ precursors contribute to conventional dendritic cells via a Zbtb46+ Ly6D+ intermediary stageREVIEWER COMMENTS

Reviewer #1 (Remarks to the Author):

Musumeci and Lutz et al. examined the heterogeneity of the CD11c+ Siglec-H+ CCR9- compartment, which was shown to contain precursors of both cDCs and pDCs. Using bulk RNA-seq, they showed that within this compartment, the B220hi (lo-hi) cells are more similar to pDCs, whereas the B220lo (lo-lo) cells are more cDC-like. Using in vitro culture assays and in vivo transfer into mice with CNS inflammation, they showed that lo-hi cells preferentially generate pDCs, while lo-lo cells seem to produce more cDCs. Next, the authors performed scRNAseq and trajectory analysis on pDCs and different DC precursor populations to dissect their developmental relationships. They showed that lo-hi cells are highly similar to the recently described 'pre-pDCs', marked by Ly6D expression [Rodriguez et al; Dress et al]. Lastly, using multi-parametric flow cytometry and in vitro cultures, they identify a Zbtb46+ Ly6D+ stage during the transition from lo-lo cells to cDCs. In addition, the authors showed that the proportional output to pDC vs cDC can be altered by IFN signalling from different precursors including lo-hi and lo-lo cells, suggesting flexibility in pDC vs cDC lineage production at late stages of DC development.

Significance:

The findings from this study provide key evidence that pDCs and cDCs are still closely related during late stages of DC development, as opposed to the recent claim by [Dress et al; Rodriguez et al] that pDC and cDC fate split very early at the myeloid vs lymphoid divergence, and that pDC represents a distinct lineage to cDC.

Major concerns:

There is an overwhelming amount of data in this manuscript and we did not feel a lot of it added to the main narrative of the story and in many cases detracted from it. We would like to suggest a different structure that might be easier to digest and deliver the core messages. There are also a lot of mistakes with regards to figures being referenced etc (see minor concerns below). We indicate key additional experiments with **. It also felt like a lot of experiments of culture etc had only been done once, as information regarding the number of independent experiments performed was not provided in the figure legends. Results should be done for the core functional or fate experiments and all data compiled.

We would like to suggest a revised structure as below:

Set the scene of your progenitors in context to others published, with bulk RNA-seq and fate analysis

Comparison of the authors populations (lolo, lohi) to the Ly6D defined progenitors of Rodriguez et alTussiwand and Dress et alGinhoux. The authors have excellent data from Supp Fig 1, and Supp figs 8-10, which we think is underutilised and aspects could be in the main figure. Through FACS-based backgating or separate plots on the tSNE UMAP, please show how much your populations overlap or are separate from those published. It doesn't matter either way but it should be clear what your starting population is relative to the others.

Introduction to new markers (B220 based separation) and gating strategy

Transcriptomics can be done, but only Fig 1 a, b, d and e is necessary for main Figure. 1c can be in supp.

Bulk fate data as per Figure 2. However, the GM-CSF, CpG and IFN data is distracting perhaps leave this out at this stage and come back to it later. The key message of pDC vs cDC bias between your lolo and lohi is lost in the current representation of the data. Instead of being in separate plots, take a progenitor centric view with different colours in columns corresponding to cDC and pDC bias. Highlight your lolo and lohi in bold colours (not grey as currently for one of them)

**The authors should demonstrate whether this fate bias exists after transfer in vivo in a non-inflammatory setting?.

Single cell analyses. The above data would provide the rationale to explore the heterogeneity of their populations in context to others. Currently Figures, 4, 5 and corresponding supplementary figures are valuable but very heavy to digest. If we are correct, the authors simply want to provide evidence that their lo-lo progenitors may be common pDC and cDC progenitors. In our opinion, just focusing on the visualisation of the structure in Figure 5, and only showing Fig 5a, b, d and only some select examples from c would be sufficient. And to overlay the sorted populations like in Fig 4a, but in the dimensionality reduction from the velocity analysis of fig 5. Also, perhaps a simpler cartoon of their proposed trajectories that are concluded from this data. In our opinion, trajectory data is hypothesis-generating not definitive demonstration. So less emphasis should be placed in describing this in detail.

Zbtb46 and separation – current Figure 6 which we believe is a key figure but could be laid out better. From this, it is not clear whether the authors progenitors are the same or different to the Ly6D based gating. The authors do see Zbtb46-expressing lolo cells freshly isolated in Fig S4a. These should be tested alongside Z- lolo and Z- lohi. The emergence of zbtb46 expression prior to cDC commitment should be a front and centre finding. We like the +IFN findings but add this as a separate new experiment (in point 5 below).

Clonal validation. **The current manuscript falls short of addressing whether the lolo population is genuinely a cDC/pDC progenitor (presumably without B cell fate), or a mixture of pDC-only and cDC-only progenitors. Really speaking, this is the core requirement to claim that previously described pre-pDCs have been misattributed. If the authors can (but not essential) do the clonal assays in medium and separately in +IFN. If the % of a common pDC/cDC fate changes to mostly pDC-only fate, that would be excellent evidence of a bipotent progenitor 'choosing' a fate based on the environment.

Emergency – We found the many experiments involving the addition of cytokine or stimuli. These findings are important but very distracting, and the CNS experiments 'out of nowhere'. If the authors wish to discuss the plasticity of their population in emergency settings, then it is perhaps best to lay the experiments out here at the end of the manuscript so they don't detract from the core message too early.

Cartoon summary of the findings. What do you want the reader to take away?

Regarding conclusions. If the authors demonstrate that prior studies were 'wrong' and that their pro-pDCs do actually generate cDCs, then the authors can be a bit braver while still being diplomatic in their conclusions. But clonal data would really be needed to support this, otherwise the argument of 'contamination' can always be made.

Some other general points:

While it seems that Ly6D+ LPs, Ly6D+Siglec-H+ LPs and lo-hi cells are highly similar transcriptionally, there is no examination of their developmental relationships in the current study. Do Ly6D+Siglec-H+ LPs largely overlap with lo-hi cells? Can Ly6D+Siglec-H+ LPs develop into lo-hi cells, or vice versa? If so, what is the kinetics, i.e. how long does it take for the transition?

Also, it is important to confirm the function of the pDC generated. A functional assay testing the cytokine production of the pDC generated from the different population should be address.

Do the lo-lo and lo-hi cells developed from CDPs (myeloid pathway), or CLPs (lymphoid pathway), or both? Also line 185, the authors claim that "comparative transcriptome analysis indicated lo-hi cells to be advanced precursors of pDCs with low proliferative potential". To conclude that from the study the other should perform a proliferation assay and also demonstrate if the lo-lo cells have the potential to differentiate into lo-hi cells.

The lo-hi cells can still generate considerable amount of cDCs in Flt3L cultures (~20%; Fig2b), suggesting that there might be heterogeneity within this population. Indeed, a few lo-hi cells are found in the pre-cDC/CDP cluster in Fig 4a. To fully characterize this population, clonal culture assays with Flt3L is required. It would also be good to perform this culture experiment side by side with the Ly6D+Siglec-H+ LPs. It is worth mentioning that a small amount of cDCs can be generated by Ly6D+Siglec-H+ LPs in the original study (Fig S2e in Rodrigues et al).

A related question: there is a small fraction of Zbtb46+ cells within the lo-hi fraction (Fig S4a), would these be the one making cDCs? This could be addressed by combining index sorting of Zbtb46-GFP BM and clonal culture assays if the authors chose.

Does lo-hi to pDC transition involve cell division? i.e. how immediate is this precursor stage? This can be answered using CTV labelling and then Flt3L culture.

The authors decided to decipher the role of inflammation on pDC development by using an EAE model. What is the relevance of that this particular model? Some may have focussed on a other acute infection models. Also, a transfer in a non-inflammatory model may have been interesting specially to argue on the authors claim line 254 "in vivo even under inflammatory condition".

Minor concerns:

For culture assays in Fig 2 & 6, only % of cDC/pDC was shown. What about cell numbers? If there was an overall increase in cell numbers, as well as unchanged or increase cDC numbers, then the authors need to re-word their conclusion from line 231-233.

In the EAE in vivo experiments, why are the transferred cells isolated from Flt3L-treated BM, but not the steady-state BM?

Fig 3b: what exactly are the controls?

Fig S3: why are pDCs not defined using CCR9 here? Even with Bst2+ sigH+, there are still CCR9-cells (i.e. not bona fide pDCs), based on Fig S1. Also, why the pDC gating strategy differ from the spleen and the CNS? What population was transferred in the representative gating strategy Fig S3.

Fig 3. The transfer of lo-lo cells gave rise to a lot of cDCs and pDCs whereas previously (Fig 2B) they clearly favor cDCs. Is it related to an in vitro/in vivo bias? An in vivo transfert without inflammation in wt mice would be nice.

Fig 3d. In the spleen, all the subsets can lostlose the expression of CD11c. Is there any explanation for that?

Line 264-267: where is the corresponding data regarding Zbtb46-GFP exclusion in Fig S4?

Fig 4a. Are the author talking about the lo-hi precursors? In the fig 4a. It seems that it is the lo-lo. Also it would have been nice but dont know if it is possible to have some cDCs.

Line 286. it seems that they may be some error in the lo-lo versus lo-hi. That sentence with the indeed just after the lo-hi make no sense for me.

Fig 4c: annotation of names to the different clusters is not clear. There are 9 clusters but only 7 populations...

Fig4e: why is the link between cluster 5 and 9 so weak, considering they looked similar based on the other UMAPs and heatmap (4d)?

Fig4f: Might be interesting to include additional plot showing B-bias score.

Line 360 – 387: this paragraph is very hard to follow, as there are few references to the exact figure panels throughout.

Fig 5: the labels in figure panel and legend are different, check and update reference in main text.

Fig 5b : The authors should precise what they mean by "typical gene". Also, the cl6 and cl0 which both contain CDP are really different in term of transcriptome. It is also interesting to observe that the cluster 2 B cell precursors is the only one with high level of Flt3 transcript.

Fig 5c: these plots for individual genes contain too much information. Is it possible to move these to supplementary and is there any way to simplify and summarize the main message in the main figure?

Line 431. Typo errors pre-cDCs written twice.

Line 434. Wrong figure cited

Fig 6b: Where are FigS8g Dress pre-pDC and S8h Leylek tDC on this UMAP? Also, the author talks about pre-cDC2 but the gating for that is not nice (well all the gating presenting in fig S8 is not good. Even the lo-lo population is not convincing).

Fig 6d: Siglec-H staining is very sub-optimal. The use of Siglec-H-BV786 and MHCII-BV650 didn't help either.

Fig S8a: IL7R staining is very sub-optimal. Compared to Fig S5, perhaps the IL7R+ gate should be adjusted for better extraction of Ly6D SP and DP.

Fig S10a: Is this UMAP showing subset 3 from S9g?

Line 474 – 478: the main conclusion is that lo-lo cells can divert into 2 routes, one to generate pDC via lo-hi stage, the other to generate cDC via Zbtb46+ Ly6D+ stage. However, this has not really been directly shown in this study.

Line 126: missing reference

Figure 1A. The gating for the preDC is not convincing. The gating on the Sirp-a is weak.

Line 158 : Why the authors refer to the S1 and S2. They have not relation within the text.

Fig 1D: the lo-lo population are few transcripts for irf5 and notch4 but present in the pDC and other subsets. Irf5 is associated with IFN response in response to viral infection. Didn't find anything relevant about notch4 but I found that interesting.

Line 183 : where does that claim come from? The Fig E do not allow to see gene signatures.

Line 185 : the author claim is strong. Because it is just based on transcriptome analysis, the utilization of the term low proliferative potential is not justified.

Fig2A-B : A representative facs plot for the lo-hi populations would be appreciated.

Also the authors claim line 203 "These results indicate that CD11c+ Siglec-H+ CCR9lo BM cells expressing high levels of B220 preferentially give rise to CCR9hi pDCs while those expressing low levels of B220 generate pDCs and cDCs. Could the lo-lo population may be contaminated with lo-hi cells I.e B220+ which are described to differentiate principally into pDC?"

Fig D. Maybe it is just me, We I found that hard to compare the wt and IFNR-/- as they are not on the same graph..

Fig S2 : a TLR9-/- may have been better to claim that ?

Line 223 . Ref missing

Line 225 and 230 : error in the figures cited. It should be Fig 2E,G

Fig 2.E. Few cDC1 are generated in medium only. Also a representative facs plot for the CpG-A or at least type I-IFN would have been nice.

Line 231 : The authors claim "Thus, exposure of CD11c+Siglec-H+CCR9lo precursors to type I IFN limited cDC output in favor of pDC output, but promoted differentiation of cDC1 and increased expression of CD86 and MHC-II". However, the panel G show CD80 and CD86 expression on cDCs derived from lo-lo precursors. The authors should rewrite that conclusion or show the related results.

Line 240. Missing ref.

Line 268. Missing ref.

Line 268. Missing ref.

Line 645. Is Ly6D really used in the Lineage?

Line 670. Typo error.

Reviewer #2 (Remarks to the Author):

In this study, Musumeci et al identified new pDC precursor populations--CD11c(+) Ly6D(+) Siglec-H(+) CCR9(lo) B220(lo) [lo-lo] precursors and more advanced CD11c(+) Ly6D(+) Siglec-H(+) CCR9(lo) B220(hi) [lo-hi] precursors--placed between previously identified CD11c(-) Ly6D(+) Siglec-H(+) precursors (pre-pDCs) and pDCs. Interestingly, lo-lo precursors still retained the potential to give rise to cDCs, probably via Siglec-H(+) Zbtb46(+) Ly6D(+) intermediary cells under inflammatory condition in vivo and under Flt3L-supplemented in vitro culture. Overall, this study is well executed, especially bioinformatics for scRNA-seq and FACS. The concept of the plasticity between pDC and cDC cell fate is an important subject. However, I have some concerns about the cell fate assays and the manuscript writing.

Major points)

1. The authors state "In conclusion, our study demonstrates that CD11c(+) Siglec-H(+) Ly6D(hi) Zbtb46(-) CCR9(lo) B220(lo) precursors retain the potential to give rise to pDCs and cDCs under steady state conditions" (line 591). However, the authors showed the generation of cDCs from lo-lo precursors only in EAE model mice and in Flt3L-supplemented in vitro cultures. On the other hand, Fig. 6A indicates that the "transitional" Zbtb46(+) Ly6D(+) Siglec-H(+) CD11c(+) cells exist in vivo in a steady state. This seems to indicate that pDC precursors such as pre-pDCs and lo-lo precursors differentiate into cDCs in a steady state in vivo. Thus, it is important to demonstrate the results of cell transfer experiments, ideally in non-irradiated normal mice, to test whether pre-pDCs and lo-lo precursors generate cDCs, via Siglec-H(+) Ly6D(+) Zbtb46(+) cells, in vivo. Also, do the in vitro experiments represent inflammatory or steady conditions?
2. The evidence that a single precursor cell can generate both pDCs and cDCs needs to be shown to draw the authors' conclusion. At least the authors should perform single cell culture experiments in vitro for lo-lo precursors (and pre-pDCs).
3. It would be important to show direct evidence of the hierarchy between the precursor populations (pre-pDCs, lo-lo precursors and lo-hi precursors) at least in vitro (ideally in vivo), not only predicting by bioinformatics.
4. Do cDCs generated from pDC precursors and those derived from CDPs have distinct functions? What would be the biological meaning of the plasticity?
5. The manuscript is very difficult to read partly due to the indefinite description of surface markers for each cell population. For example, lo-lo precursors are indicated as CD11c(+) MHCII(-/lo) Siglec-H(+) CCR9(lo) pDC-like cells, B220(lo) Ly6D(+) Zbtb46(-) cells, CCR9(lo) B220(lo) cells, CD11c(+) Ly6D(+) Siglec-H(+) CCR9(lo) B220(lo) cells, and so on. Also, I recommend shortening the manuscript, especially the bioinformatics part, to focus on the essential points.

Minor points)

1. Lines 196 to 203: The statements "the lo-lo precursors generated mainly cDCs and few CCR9(hi) pDCs" and "those expressing low levels of B220 generate pDCs and cDCs" appear to be conflicting.
2. Lines 225, 229, and 230: Fig numbers are incorrect (should be Fig. 2).
3. Fig. S4b: Are the lo-lo cells in this Fig a Zbtb46(-) cell fraction as stated in the main text (line 264)? If so, please indicate so in the Fig. This also applies to Fig. 6.
4. Line 276: Are Ly6D(+) LP a Siglec-H(+) fraction of LPs? If so, please indicate so in the Fig. This also applies to Fig. 6. It would be better to state the definition of LPs in the text.
5. Fig. 5: Where are lo-lo precursors projected in this RNA velocity assay?
6. Fig. 6A: "/" within "Lin(-) Flt3(-) CD11c(+) B220(-) Siglec-H(-)/p1" is unclear. Does this "/" mean "and"? Because "Lin(-) Flt3(-) CD11c(+) B220(-) Siglec-H(-)" is identical to the p1 in Fig. 1, it is confusing. Does "Siglec-H(-)/(+) pre-cDC" mean "Siglec-H(-) or Siglec-H(+) pre-cDC" ?

I have attached a PDF file in which superscript letters are used.

Reviewer #3 (Remarks to the Author):

The main claim of the paper is that there is plasticity between the lineages of plasmacytoid dendritic cells (pDCs) and conventional dendritic cells (cDCs). Specifically, the author identified a bone marrow (BM) precursor stage coined "lo-lo" that is already engaged in differentiation towards pDCs but can still give rise to cDCs, and whose fate is influenced by the inflammatory milieu since type I interferon (IFN-I) promoted its differentiation towards pDCs over cDCs in vitro. This proximal BM precursor of pDCs is phenotypically characterized as Lineage(CD3, CD19, Ly6G, NK1.1, Ter119)-neg CD135-high CD11c-pos Siglec-H-pos CCR9-low B220-low Ly6D-high Zbtb46-neg. It is also Ly6C-neg, CD115-pos and CXCR4-pos.

Moreover, by combining single cell RNA-sequencing (scRNA-seq), high content flow cytometry, and in vitro differentiation assays with kinetic measurement of key lineage-marker genes by qRT-PCR, the authors mapped the differentiation trajectory of this "lo-lo" precursor into pDCs and cDCs. They showed that this "lo-lo" precursor gave rise to cDCs by going through a transient state co-expressing Ly6D and Zbtb46. Immediately following ex vivo purification from BM, this Ly6D-pos Zbtb46-pos transient state expressed key pDC-lineage marker genes including Tcf4, Spi1 and Irf8. However, it lost the expression of these pDC-lineage marker genes under the differentiation conditions used by the authors in vitro while inducing key cDC-lineage marker genes including Batf3, Id2 and Spi1.

The "lo-lo" precursor characterized and studied by the authors expresses CD11c. Hence, it is different from the BM precursor shared between pDCs and B cells that has been previously identified as also Siglec-H-pos Ly6D-high but CD11c-neg ("Ly6D-pos lymphoid progenitor (LP)", manuscript references 37 & 38). Most importantly, the authors convincingly showed that the "lo-lo" progenitor gave rise to both pDCs and cDCs, contrary to the "Ly6D-pos LP" that was reported not to yield cDCs but exclusively B cells and pDCs (ref. 37 & 38).

This study is original and convincing. The results support rather well the conclusions drawn. It should be of great interest for the researchers interested in the ontogeny and functions of pDCs and cDCs, since it brings novel and important information regarding the plasticity of these two lineages of immune cells and their relationships. This is timely considering the ongoing controversy on the identity of pDCs, which have been proposed recently not to belong to the family of DCs but to that of innate lymphoid cells (ref. 37). The data in this manuscript show that proximal pDC progenitors are plastic and can differentiate into cDC, showing that there exist an ontogeny proximity between pDC and cDC. Importantly, these results are consistent with the reciprocal molecular regulation of the differentiation of pDCs versus type 1 cDCs by opposing combinations of a handful of master transcription factors whose artificial perturbation can lead to the reprogramming of pDCs into cells resembling cDC1s or conversely. This is discussed well by the authors (lines 74-78, and 525-531, manuscript references 31-35, 61, 62). Hence, this study should become a key component of the scientific corpus advancing our understanding of pDC identity and of their relationship to cDCs.

Two major points and a few minor points need to be addressed.

Major points.

1) Alternative explanations for the effects of IFN-I on the output of in vitro differentiation cultures. The output of in vitro cultures are given as percent of pDCs or cDCs within CD11c-pos cells (Figures 2c-d, 2f; 6c-d). Neither absolute numbers of DC subsets nor global cell yields of these cultures are given. This is a major issue, because IFN-I have anti-proliferative and pro-apoptotic

functions differentially affecting distinct cell types. Hence, IFN-I could induce the preferential apoptosis of the cells engaged in cDC differentiation, and/or inhibit their proliferation, while sparing the cells engaged into pDC differentiation. Such a mechanism of differential selection acting on a population of heterogeneous precursors already committed to either the pDC or cDC lineage could explain the strong relative enrichment observed in the output of pDCs in the cultures. Hence, one cannot exclude alternative explanations to the interpretation given by the authors that IFN-I instructs the fate choice of individual bipotent BM precursors towards pDCs over cDCs. The authors must address this issue experimentally, by providing absolute numbers of cDCs and pDCs retrieved per well for each culture condition, or at least the total yield of cells per well. For each input precursor population, they could also measure cell apoptosis and proliferation by flow cytometry kinetically over time comparing medium alone to IFN-I.

2) In vivo fate plasticity of the Ly6D-high Zbtb46-neg "lo-lo" precursors.

2-1) The experiment of the in vivo cell fate of transferred "lo-lo" precursors shown in figure 3 is not conclusive, because this population is heterogeneous and encompasses a significant fraction of Ly6C-pos Zbtb46-pos pre-cDCs, as shown later in the manuscript. Hence, to rigorously demonstrate that the "lo-lo" precursor cells can give rise to both pDCs and cDCs in vivo, the authors need to repeat that experiment using the strategy designed later in the paper to remove the contaminating pre-cDCs, namely sorting Ly6D-high Zbtb46-neg "lo-lo" precursors.

2-2) The data from figure 3 show that, upon in vivo adoptive transfer, the "lo-lo" precursors gave preferentially rise to pDCs (~55% of CD45.1-pos cells) over cDCs (~30%). This is strikingly different from what was observed with the in vitro differentiation assay, as shown on figure 2c-d where the output was on the contrary strongly biased towards cDCs (~80%) over pDCs (~10%). It is not clear whether this discrepancy reflects major differences in the differentiation process as it occurs in vivo as opposed to in vitro, or whether it is due to the inflammatory condition associated to the EAE model. Hence, the authors must compare the in vivo differentiation of the Ly6D-high Zbtb46-neg "lo-lo" precursors upon their adoptive transfer not only in animals induced to develop EAE but also in control animals at ground state (no inflammation). Ideally, the authors should also include recipient animals injected by CpG or infected with a virus, with adoptive transfer of the Ly6D-high Zbtb46-neg "lo-lo" precursors at the time of the peak production of IFN-I in vivo, to better mirror the inflammation conditions used in vitro.

Minor points.

a) Please give additional information to allow the reader comparing more precisely the complementary analyses of the scRNA-seq data from Figure 4 & 5. Figure 4 & 5 correspond to different, complementary, analyses of the same scRNA-seq dataset, and the clustering of cells were performed independently with distinct methods for each analysis (RaceID yielding 8 clusters in Figure 4, versus Louvain from scvelo yielding 7 clusters in Figure 5). It is not clear if the cells identified as monocytes in Figure 4 (cluster 7) are still included in the analysis of Figure 5, and if yes, to which cluster do they belong. Does cluster 9 from Figure 4 correspond to Ly6D-pos LP and hence to cluster 4 from Figure 5? Does cluster 4 from Figure 4 correspond to cluster 3 from Figure 5? To help the reader comparing these analyses, the authors should provide a table/heatmap indicating the % overlap in the individual cell content of each (Figure 4/Figure 5) cluster pair.

b) Please discuss the discrepancy between your study and those on the Ly6D-pos Siglec-H-pos LP. As the authors proposed, it is likely that the CD11c-pos "lo-lo" precursor they characterized is just downstream of the CD11c-neg Ly6D-pos Siglec-H-pos LP previously identified and characterized by others (ref. 37 & 38). In that case, the fact that the CD11c-neg Ly6D-pos Siglec-H-pos LP was reported to give rise only to B cells and pDCs but not to cDCs is inconsistent with the authors' results showing that the CD11c-pos "lo-lo" precursor gave rise to both pDCs and cDCs. Could the authors please discuss this discrepancy?

c) Please italicize the names of genes throughout the paper, both in the text and on the figures.

d) Please carefully revise and correct figure and table calling in the main text and in figure

legends.

There are mistakes in figure and table calling in the main text and in figure legends. A non-exhaustive list of example is provided thereafter. In the legend of figure 5 panel a, the text "(same data as in Fig. 5)" must be corrected as "(same data as in Fig. 4)". Line 225, "(Fig. 1E, G)" must be replaced by "(Fig. 2E, G)". Line 229, "(Fig. 1F)" must be replaced by "(Fig. 2F)". Line 230, "(Fig. 1G)" must be replaced by "(Fig. 2G)". Line 645, replace "(see Supplemental Table 1)" by "(see Table S3)", etc...

e) Line 645, there is a mistake in the list of antibodies used in the lineage-depletion cocktail: replace "Ly6D" by "Ly6G".

f) Literature citation.

The authors have covered the relevant literature accurately and rather extensively. However, when mentioning the specific finding that "DC subpopulations are conserved across species" line 50, they should cite the first result papers that demonstrated it for the first time (PMID: 18218067; 20193019). Alternately, they could cite a relatively recent review specifically discussing this concept, from authors who put it forward by comparing mouse and human DCs and then extended it across several warm blooded vertebrate species (e.g. PMID: 26082777).

Point-by point Reply to the reviewers' comments

We thank the reviewers for their valuable feedback that has greatly improved the manuscript.

Reviewer #1 (Remarks to the Author):

Musumeci and Lutz et al. examined the heterogeneity of the CD11c⁺ Siglec-H⁺ CCR9⁻ compartment, which was shown to contain precursors of both cDCs and pDCs. Using bulk RNA-seq, they showed that within this compartment, the B220^{hi} (lo-hi) cells are more similar to pDCs, whereas the B220^{lo} (lo-lo) cells are more cDC-like. Using in vitro culture assays and in vivo transfer into mice with CNS inflammation, they showed that lo-hi cells preferentially generate pDCs, while lo-lo cells seem to produce more cDCs. Next, the authors performed scRNAseq and trajectory analysis on pDCs and different DC precursor populations to dissect their developmental relationships. They showed that lo-hi cells are highly similar to the recently described 'pre-pDCs', marked by Ly6D expression [Rodriguez et al; Dress et al]. Lastly, using multi-parametric flow cytometry and in vitro cultures, they identify a Zbtb46⁺ Ly6D⁺ stage during the transition from lo-lo cells to cDCs. In addition, the authors showed that the proportional output to pDC vs cDC can be altered by IFN signalling from different precursors including lo-hi and lo-lo cells, suggesting flexibility in pDC vs cDC lineage production at late stages of DC development.

Significance:

The findings from this study provide key evidence that pDCs and cDCs are still closely related during late stages of DC development, as opposed to the recent claim by [Dress et al; Rodriguez et al] that pDC and cDC fate split very early at the myeloid vs lymphoid divergence, and that pDC represents a distinct lineage to cDC.

Major concerns:

1. There is an overwhelming amount of data in this manuscript and we did not feel a lot of it added to the main narrative of the story and in many cases detracted from it. We would like to suggest a different structure that might be easier to digest and deliver the core messages. There are also a lot of mistakes with regards to figures being referenced etc (see minor concerns below). We indicate key additional experiments with **. It also felt like a lot of experiments of culture etc had only been done once, as information regarding the number of independent experiments performed was not provided in the figure legends. Results should be done for the core functional or fate experiments and all data compiled.

We would like to suggest a revised structure as below:

To 1. We thank the reviewer for the time and effort that went into reviewing our manuscript and providing valuable and detailed feedback to improve it. As suggested, we have restructured the narrative and have also shortened the manuscript adopting most of the suggested changes in the revised manuscript.

Specifically, we have moved the bulk RNAseq clustering analysis (former Fig. 1c) to the supplement (revised Fig. S2), integrated the results of differentiation assays

without stimulation (former Fig. 2) into revised Fig. 1, have taken out the results from adoptive transfer into mice with EAE (former Fig. 3) and condensed the scRNAseq data into one Figure (revised Fig. 2) leaving out the former Fig. 4 which was felt to be redundant. A comprehensive phenotype comparison of the precursor subsets described here and in other publications is now shown in revised Fig. 3. New data showing in vivo cell fate of the precursors confirming in vitro cell fate results were added in revised Fig. 5 and new data on the effect of type I IFN on pDC vs cDC output from the precursors subsets isolated using the refined gating strategy are shown in revised Fig. 6 at the end as suggested. Clonal assays were attempted, but failed due to the low output of these advanced precursors. New data showing detection of cell division with CellTrace Blue labeling were included in revised Fig. 4 and 6 to address the question of dual potential versus precommitment.

Information regarding the number of independent experiments are provided in the legends. The individual data points are shown in the graphs.

2. Set the scene of your progenitors in context to others published, with bulk RNA-seq and fate analysis

3. Comparison of the authors populations (lolo, lohi) to the Ly6D defined progenitors of Rodriguez et al. Tussiwand and Dress et al. Ginhoux. The authors have excellent data from Supp Fig 1, and Supp figs 8-10, which we think is underutilised and aspects could be in the main figure. Through FACS-based backgating or separate plots on the tSNE UMAP, please show how much your populations overlap or are separate from those published. It doesn't matter either way but it should be clear what your starting population is relative to the others.

To 2 and 3: To put the precursor populations into context with the pDC precursor populations investigated by the two other studies as suggested, we show a comprehensive phenotypic analysis in revised Fig. 3. This figure shows the results of our high-dimensional flow cytometric analysis indicating the different populations in the UMAP.

We additionally revised the text throughout the manuscript to clarify how the populations differ: The Ly6D⁺ Siglec-H⁺ LP giving rise to pDCs (described by Rodrigues et al.) and the pre-pDC defined by Dress et al. are CD11c-negative. These populations do not overlap with the lo-lo or lo-hi precursors described by us, which express CD11c and lower levels of CD135 and higher level of Siglec-H. There are additional differences in the expression of some other markers, including CD135 (shown in revised Fig. 3). This is also addressed in the discussion (page 21, lines 546-548).

4. Introduction to new markers (B220 based separation) and gating strategy

5. Transcriptomics can be done, but only Fig 1 a, b, d and e is necessary for main Figure. 1c can be in supp.

To 4 and 5: Initial gating strategy and separation based on B220 as well as bulk RNAseq data of the four populations are shown in Fig. 1 as before. Former Fig. 1c was moved to the supplement (revised Fig. S2), the results text was shortened.

6. Bulk fate data as per Figure 2. However, the GM-CSF, CpG and IFN data is distracting perhaps leave this out at this stage and come back to it later. The key message of pDC vs cDC bias between your lolo and lohi is lost in the current representation of the data. Instead of being in separate plots, take a progenitor centric view with different colours in columns corresponding to cDC and pDC bias. Highlight your lolo and lohi in bold colours (not grey as currently for one of them)

To 6: As recommended by the reviewer we integrated the cell fate assay results from former Fig. 2b and c into revised Fig. 1. Here we show only the steady state condition (wildtype, Flt3L) combining the data from former Fig. 2b and c. cDC and pDC output are shown in one plot for direct comparison of the cell fates of the precursor populations as suggested. As recommended we show the IFN I effect at the end of the manuscript (new data using the refined sort strategy) in revised Figure 6. The CpG and IFN effects on IFNAR-KO vs WT precursors from former Fig. 2b, c, d are shown in the supplement (revised Figure S9). Data in former Figures 2a, b, e, f, g were omitted from the revised manuscript for sake of clarity and conciseness.

7. **The authors should demonstrate whether this fate bias exists after transfer in vivo in a non-inflammatory setting?.

As recommended adoptive transfers of Ly6D^{hi} Zbtb46⁻ lo-lo and lo-hi precursors into untreated non-irradiated mice were performed and the progeny were analysed by multiparameter flow cytometry in the spleen showing generation of pDCs, cDC2 and few cDC1 as well as some Zbtb46⁺ Ly6D⁺ cells after 5 days. For comparison Zbtb46⁺ Ly6D⁺ intermediary cells (isolated from BM cells) were also transferred and generation of cDCs was observed after 3 days (shown in revised Fig. 5).

8. Single cell analyses. The above data would provide the rationale to explore the heterogeneity of their populations in context to others. Currently Figures, 4, 5 and corresponding supplementary figures are valuable but very heavy to digest. If we are correct, the authors simply want to provide evidence that their lo-lo progenitors may be common pDC and cDC progenitors. In our opinion, just focusing on the visualisation of the structure in Figure 5, and only showing Fig 5a, b, d and only some select examples from c would be sufficient. And to overlay the sorted populations like in Fig 4a, but in the dimensionality reduction from the velocity analysis of fig 5. Also, perhaps a simpler cartoon of their proposed trajectories that are concluded from this data. In our opinion, trajectory data is hypothesis-generating not definitive demonstration. So less emphasis should be placed in describing this in detail.

To 8: As suggested we shortened this part. The RACE-ID and FATE-ID analyses of the scRNA-seq data set (shown in former Fig. 4) were omitted and only selected velocity plots are now shown in the revised Fig. 2. High-dimensional flow cytometric analysis (shown in revised Fig. 3) provides further evidence for the connection between pDC- and cDC-primed precursors and cell fate data shown in revised Fig. 4 show that Ly6D^{hi} Zbtb46⁻ lo-lo and lo-hi precursors can generate cDCs, but pre-cDC subsets do not generate pDCs in steady state conditions.

9. Zbtb46 and separation – current Figure 6 which we believe is a key figure but could be laid out better. From this, it is not clear whether the authors progenitors are the same or different to the Ly6D based gating. The authors do see Zbtb46-expressing lolo cells freshly isolated in Fig S4a. These should be tested alongside Z–

lolo and Z⁻ lohi. The emergence of zbtbt46 expression prior to cDC commitment should be a front and centre finding. We like the +IFN findings but add this as a separate new experiment (in point 5 below).

To 9: We changed the results text to make it clear that the lo-lo cells, lo-hi cells and pDCs were strictly sorted as Ly6D^{hi} Zbtb46⁻ excluding any Zbtb46⁺ cells. This refined gating strategy was used for multidimensional phenotyping shown in revised Fig. 3 (p. 11, line 318-320) and for cell fate assays shown in Figure 4, 5 and 6 (p. 13, line 364-365). To make this clear we designate these populations as Ly6D^{hi} Zbtb46⁻ lo-lo and lo-hi precursors.

As explained above (reply to points 2 and 3) these Ly6D^{hi} Zbtb46⁻ lo-lo and lo-hi precursors were distinct from Ly6D⁺ Siglec-H⁺ LP (Rodrigues et al.). The emergence of Zbtb46 expression was already observed after 20 hrs of culture of the Ly6D^{hi} Zbtb46⁻ lo-lo cells (see revised Fig. 4 h).

Precursor fractions containing Zbtb46-expressing cells (Siglec-H⁺- pre-cDCs, Siglec-H⁻ pre-cDCs and Zbtb46⁺ Ly6D⁺ intermediate cells) were tested alongside the other populations in cell fate assays and showed cDC-commitment consistent with the literature (see revised Fig. 4 a).

10. Clonal validation. **The current manuscript falls short of addressing whether the lolo population is genuinely a cDC/pDC progenitor (presumably without B cell fate), or a mixture of pDC-only and cDC-only progenitors. Really speaking, this is the core requirement to claim that previously described pre-pDCs have been misattributed. If the authors can (but not essential) do the clonal assays in medium and separately in +IFN. If the % of a common pDC/cDC fate changes to mostly pDC-only fate, that would be excellent evidence of a bipotent progenitor 'choosing' a fate based on the environment.

To 10: We attempted to do the suggested clonal assays, but due to the low number of divisions couldn't detect enough cells to assess the cell fate of individual precursors.

Ly6D^{hi} Zbtb46⁻ lo-lo precursors generated pDCs and cDCs (revised Fig. 4a, b). To investigate whether these precursors have dual pDC/cDC potential or whether they contain pDC- and cDC-primed cells we followed their division and phenotypic changes by Cell Trace Blue labeling that was done before sorting the subsets in order to obtain comparable labeling. As described on p. 14, line 397-406 and shown in revised Fig. 4f upregulation of Zbtb46 and MHC II as well reduced expression of Ly6D and Siglec-H in the undivided fraction after 3 days of culture indicated that a part of the Ly6D^{hi} Zbtb46⁻ lo-lo precursors had started to differentiate into cDCs before the first division and subsequently expanded. Zbtb46 expression emerged early after 20 hrs of culture before cell division (p. 15, line 406-409; revised Fig. 4 g and h). Addition of IFN- α to the culture inhibited cell division and arrested cDC-primed cells in the Zbtb46⁺ Ly6D⁺ state, while at the same time promoting pDC differentiation (p. 17, line 471-475; revised Fig. 6). We conclude that "cell fate decision before division argues against dual potential and shows that the Ly6D^{hi} Zbtb46⁻ lo-lo precursor fraction contains cDC-primed cells in addition to pDC precursors." (p.15, 409-411). We included a graphical display of the two models, dual potential vs precommitment in revised Fig. S9 and describe our interpretation of these results in the discussion (p.19, line 494-503).

11. Emergency – We found the many experiments involving the addition of cytokine or stimuli. These findings are important but very distracting, and the CNS experiments ‘out of nowhere’. If the authors wish to discuss the plasticity of their population in emergency settings, then it is perhaps best to lay the experiments out here at the end of the manuscript so they don’t detract from the core message too early.

To 11: Based on the reviewer’s comments we decided to remove the cell transfer experiment into mice with ongoing EAE and also the in vitro differentiation in presence of GM-CSF. The effect of IFN- α on pDC vs cDC output is shown at the end in revised Fig. 6.

12. Cartoon summary of the findings. What do you want the reader to take away?

13. Regarding conclusions. If the authors demonstrate that prior studies were ‘wrong’ and that their pro-pDCs do actually generate cDCs, then the authors can be a bit braver while still being diplomatic in their conclusions. But clonal data would really be needed to support this, otherwise the argument of ‘contamination’ can always be made.

To 12 and 13: We thank the reviewer for this suggestion. We have added a cartoon describing the two models, how the Ly6D^{hi} Zbtb46⁻ lo-lo precursors could generate cDCs and pDCs (revised Fig. S9). The model supported by our data is described in the discussion (p. 19, line 494-503).

We can rule out that contamination with Zbtb46⁺ pre-cDC led to the generation of cDCs in the culture of lo-lo cells, because the lo-lo cells used for cell fate assays in vitro and in vivo (revised Fig. 4 and 5) were strictly sorted as Ly6D^{hi} Zbtb46⁻ and showed de novo expression of Zbtb46-eGFP in undivided cells (revised Fig. 4f, g). This is discussed in the revised manuscript (p.19, line 503 – 506). Thus, “high level expression of Ly6D Siglec-H⁺ precursor cells does not exclude cDC cell fate” (p.14, line 380-381).

Similar results were obtained with the Ly6D⁺ Siglec-H⁺ LP, which generated pDCs and cDCs after culture with Flt3L. The frequency of pDCs was higher at early time points whereas the frequency and total number of cDCs was higher at later timepoints (see below and revised Fig. S8). This added data is described on p. 15, line 413-426 and discussed on p. 21, line 546-552).

Fig. 1 Cell fate of Ly6D⁺ Siglec H⁺ LP in culture with Flt3L

Ly6D⁺ Siglec H⁺ LP were from Zbtb46-eGFP BM cells sorted as described by Rodrigues et al. (Lin⁻ CD16/32⁻ B220⁻ Ly6C⁻ CD135⁺ CD117^{lo/int} CD115⁻ IL7R⁺ Ly6D⁺ Siglec H⁺, see Figure S8), and cultured with Flt3L (w/o stromal cells) for 1 to 4 days. The phenotype of the progeny was evaluated by flow cytometry and is shown in (a) as percentage of cells with the respective phenotype within living cells and in (b) as number of cells with the respective phenotype indicated by colours.

It was shown in the supplemental data of the paper by Rodrigues et al. that Ly6D⁺ Siglec-H⁺ LP generated approximately 10 % cDCs in culture with Flt3L and 20 % cDC in culture with Flt3L on OP9 stromal cells. We discuss this in comparison with our own observations in the revised manuscript (p. 21, line 552-556).

Some other general points:

14. While it seems that Ly6D⁺ LPs, Ly6D⁺Siglec-H⁺ LPs and lo-hi cells are highly similar transcriptionally, there is no examination of their developmental relationships in the current study. Do Ly6D⁺Siglec-H⁺ LPs largely overlap with lo-hi cells? Can Ly6D⁺Siglec-H⁺ LPs develop into lo-hi cells, or vice versa? If so, what is the kinetics, i.e. how long does it take for the transition?

To 14: Ly6D⁺Siglec-H⁺ LPs do not overlap with lo-hi cells, because they express neither CD11c nor B220. We have included new data showing that Ly6D⁺ Siglec-H⁺ LP (DP) generated lo-hi cells within 24 hours of culture. (revised Fig. S8, results text p. 15, line 413-426). Lo-hi cells largely acquired the phenotype of differentiated pDCs in vitro and after transfer in vivo (revised Fig. 4a and 5b, p.13, line 373, p. 16, line 435-436). Reversion of the lo-hi phenotype (loss of B220 and CD11c expression) was not observed.

15. Also, it is important to confirm the function of the pDC generated. A functional assay testing the cytokine production of the pDC generated from the different population should be address.

To 15: As suggested, we stimulated the progeny of the Ly6D^{hi} Zbtb46^{-lo-lo} and lo-hi precursors with CpG-A, which stimulates IFN- α production exclusively in pDCs. In both conditions we detected substantial concentrations of IFN- α in the supernatants (revised Fig. S7, p. 13, line 375-377).

16. Do the lo-lo and lo-hi cells developed from CDPs (myeloid pathway), or CLPs (lymphoid pathway), or both? Also line 185, the authors claim that "comparative transcriptome analysis indicated lo-hi cells to be advanced precursors of pDCs with low proliferative potential". To conclude that from the study the other should perform a proliferation assay and also demonstrate if the lo-lo cells have the potential to differentiate into lo-hi cells.

To 16: To address these points, additional data is shown in the supplement of the revised manuscript to address the origin of lo-lo and lo-hi cells (shown in revised Fig. S8, described on p. 15, line 413-426, discussed on p.21, line 548-559). CDP (sorted as Lin⁻ B220⁻ Ly6C⁻ CD135⁺ CD117^{lo-int} CD115⁺ IL7R⁻ CD11c⁻ MHCII⁻) did not generate lo-lo cells lo-hi cells or pDCs culture with Flt3L on stromal cells. Ly6D⁺ Siglec-H⁺ LP (DP, sorted as Lin⁻ CD16/32⁻ B220⁻ Ly6C⁻ CD135⁺ CD117^{lo-int} CD115⁻ IL7R⁺ Ly6D⁺ Siglec-H⁺) generated lo-hi precursors, pDCs, Zbtb46⁺ Ly6D⁺ intermediary cells, cDCs and few but detectable lo-lo precursors.

Additional data on proliferation of Ly6D^{hi} Zbtb46^{lo-lo} and lo-hi cells during culture with Flt3L was included in revised Fig. 4 and quantification is also shown in revised Fig. S9. The highest proliferation was seen in pre-cDCs, followed by Zbtb46⁺ Ly6D⁺ cells and lo-lo precursors. The lo-hi cells barely proliferated in line with an advanced precursor state. The lo-lo cells also generated lo-hi cells (see revised Fig. 1 e). This is mentioned in the revised results section (p. 6/7, line 201-204).

17. The lo-hi cells can still generate considerable amount of cDCs in Flt3L cultures (~20%; Fig2b), suggesting that there might be heterogeneity within this population. Indeed, a few lo-hi cells are found in the pre-cDC/CDP cluster in Fig 4a. To fully characterize this population, clonal culture assays with Flt3L is required. It would also be good to perform this culture experiment side by side with the Ly6D⁺Siglec-H⁺ LPs. It is worth mentioning that a small amount of cDCs can be generated by Ly6D⁺Siglec-H⁺ LPs in the original study (Fig S2e in Rodrigues et al).

To 17: It is true that we still saw some cDC generation from lo-hi cells in vitro with variability between experiments. This is likely due to the fact that B220 shows continuous expression from low to high making the separation of lo-lo and lo-hi cells difficult. Inclusion of a small number of cDC-primed lo-lo cells in the lo-hi fraction could lead to remaining cDC output that was observed. The cell fate of transferred lo-hi cells in vivo was almost exclusively pDCs (revised Fig. 5). The difference between in vitro and in vivo could be due to reduced expansion of cDC-primed cells and better survival of pDCs in vivo than in vitro.

The cDC output from Ly6D⁺ Siglec-H⁺ LPs that we observed (revised Fig. S8) and that was shown in supplemental figure S2e of Rodrigues et al. is discussed in the revised manuscript on p. 21, line 554-556 (see answer to point 13).

18. A related question: there is a small fraction of Zbtb46⁺ cells within the lo-hi fraction (Fig S4a), would these be the one making cDCs? This could be addressed by combining index sorting of Zbtb46-GFP BM and clonal culture assays if the authors chose.

To 18: We apologize for not being clearer on this. In former Fig. S4a lo-hi cells were gated using the initial gating strategy without prior exclusion of Zbtb46, whereas in former Fig. 6 showing results of in vitro differentiation assays (now revised Fig. 4) lo-hi precursors were sorted as Ly6D^{hi} Zbtb46⁻ and did not contain Zbtb46⁺ cells (see answer to point 9). The same was true for results of cell transfer experiments shown in revised Fig. 5). This is more clearly stated in the revised manuscript (p.13, line 364-365).

19. Does lo-hi to pDC transition involve cell division? i.e. how immediate is this precursor stage? This can be answered using CTB labelling and then Flt3L culture.

To 19: As suggested we performed CTB-labelling and confirmed that cells derived from lo-hi precursors in vitro and in vivo, which showed the phenotype of differentiated pDCs, had barely proliferated showing a similar CTB profile as CCR9^{hi} pDCs sorted directly from BM cells (revised Fig. 4e, 5e, S9f). Also, in the scRNAseq data and the high-dimensional flow cytometry results we saw that the lo-hi precursors are very similar to pDCs consistent with an immediate precursor state (revised Fig. 2 and 3, p. 10, line 292).

20. The authors decided to decipher the role of inflammation on pDC development by using an EAE model. What is the relevance of that this particular model? Some may have focussed on a other acute infection models. Also, a transfer in a non-inflammatory model may have been interesting specially to argue on the authors claim line 254 “in vivo even under inflammatory condition”.

To 20: As stated above (answer to point 11) we followed the recommendation to leave out this data. In vivo data showing cell fate after adoptive transfer of lo-lo and lo-hi precursors into untreated non-irradiated wildtype mice was added in revised Fig. 5.

Minor concerns:

21. For culture assays in Fig 2 & 6, only % of cDC/pDC was shown. What about cell numbers? If there was an overall increase in cell numbers, as well as unchanged or increase cDC numbers, then the authors need to re-word their conclusion from line 231-233.

To 21: Cell output numbers (normalized to input) are now shown in the supplementary Figure S7 and S9 and mentioned in the results text.

22. In the EAE in vivo experiments, why are the transferred cells isolated from Flt3L-treated BM, but not the steady-state BM?

23. Fig 3b: what exactly are the controls?

24. Fig S3: why are pDCs not defined using CCR9 here? Even with Bst2+ sigH+, there are still CCR9- cells (i.e. not bona fide pDCs), based on Fig S1. Also, why the pDC gating strategy differ from the spleen and the CNS? What population was transferred in the representative gating strategy Fig S3.

25. Fig 3. The transfer of lo-lo cells gave rise to a lot of cDCs and pDCs whereas previously (Fig 2B) they clearly favor cDCs. Is it related to an in vitro/in vivo bias? An in vivo transfert without inflammation in wt mice would be nice.

26. Fig 3d. In the spleen, all the subsets can lostlose the expression of CD11c. Is there any explanation for that?

To 22 - 36: The data from the cell transfer into mice with EAE were removed from the revised manuscript as stated above, because they do not influence the message of the paper.

27. Line 264-267: where is the corresponding data regarding Zbtb46-GFP exclusion in Fig S4?

To 27: Former Fig. S4 was omitted, as it was redundant with the data shown in revised Fig. 4, where the refined gating strategy (exclusion of Zbtb46+ cells and gating on Ly6D^{hi}) was used.

28. Fig 4a. Are the author talking about the lo-hi precursors? In the fig 4a. It seems that it is the lo-lo. Also it would have been nice but dont know if it is possible to have

some cDCs.

29. Line 286. it seems that they may be some error in the lo-lo versus lo-hi. That sentence with the indeed just after the lo-hi make no sense for me.

30. Fig 4c: annotation of names to the different clusters is not clear. There are 9 clusters but only 7 populations...

31. Fig4e: why is the link between cluster 5 and 9 so weak, considering they looked similar based on the other UMAPs and heatmap (4d)?

32. Fig4f: Might be interesting to include additional plot showing B-bias score.

To 28 - 32: The former Fig. 4 containing RACE-ID analysis of the scRNA-seq data set and the corresponding results text were omitted per recommendation of the reviewer. cDCs were not included in the scRNA-seq experiment, but were included in the high dimensional flow cytometric analysis shown in revised Fig. 3.

33. Line 360 – 387: this paragraph is very hard to follow, as there are few references to the exact figure panels throughout.

To 33: Thank you for pointing this out, we added the references to the figure panels to make it easier to follow.

34. Fig 5: the labels in figure panel and legend are different, check and update reference in main text.

To 34: This was corrected.

35. Fig 5b : The authors should precise what they mean by “typical gene”. Also, the cl6 and cl0 which both contain CDP are really different in term of transcriptome. It is also interesting to observe that the cluster 2 B cell precursors is the only one with high level of Flt3 transcript.

To 35: The results text has been adapted accordingly.

36. Fig 5c: these plots for individual genes contain too much information. Is it possible to move these to supplementary and is there any way to simplify and summarize the main message in the main figure?

To 36: We now show only selected velocity plots in the revised Fig. 2 and moved the other plots to the supplement.

37. Line 431. Typo errors pre-cDCs written twice.

38. Line 434. Wrong figure cited

To 37, 38: The corrections have been made in the text.

39. Fig 6b: Where are FigS8g Dress pre-pDC and S8h Leylek tDC on this UMAP?

Also, the author talks about pre-cDC2 but the gating for that is not nice (well all the gating presenting in fig S8 is not good. Even the lo-lo population is not convincing).

To 39: We have further optimized the staining panel as suggested and repeated this experiment. The results were confirmed. The gating using the optimized panel is shown in the revised Fig. S5. Dress pre-pDC are shown in the UMAP in revised Fig. 3 d (2, light green) – they are encompassed in the Ly6D⁺ Siglec H⁺ LP population. The Leylek tDCs (CD11c^{low} and CD11c^{high}) have been gated in the spleen and are shown in the concatenated UMAP in revised Fig. 3 d. Leylek et al. reported that tDCs were not detectable in the BM. We found cells resembling the Leylek CD11c^{low} tDCs also in BM and these partially overlapped with the Zbtb46⁺ Ly6D⁺ intermediary cells. These are also shown in the BM UMAP in revised Fig. S6. These results are described in detail on p. 11-13, line 322-357.

40. Fig 6d: Siglec-H staining is very sub-optimal. The use of Siglec-H-BV786 and MHCII-BV650 didn't help either.

To 40: In repeat experiments the staining was optimized (see revised Fig. 4 b). Similar results were obtained and summarized with the previous results in revised Fig. 4 a.

41. Fig S8a: IL7R staining is very sub-optimal. Compared to Fig S5, perhaps the IL7R⁺ gate should be adjusted for better extraction of Ly6D SP and DP.

To 41: This has been optimized as shown below and in revised Figure S5 to obtain better gating of the Ly6D⁺ SP and DP. IL-7R⁺ cells can be clearly identified within the CD117^{lo/int} CD135⁺ population and are clearly separated by Ly6D vs Siglec H.

Fig. 2 Gating of Ly6D⁺ Siglec H⁻ and Ly6D⁺ Siglec H⁺ LP (SP and DP)
(copied from Figure S5a of the revised manuscript)

42. Fig S10a: Is this UMAP showing subset 3 from S9g?

To 42: The high-dimensional flow cytometry experiments were repeated with the optimized panel and the UMAP analysis was redone. We describe the analysis steps in the legend to revised Fig. 3. The figure showing the prior gating and exclusion steps of the UMAP analysis was removed to reduce the number of supplementary figures. We can put it in if requested by the reviewers.

43. Line 474 – 478: the main conclusion is that lo-lo cells can divert into 2 routes, one to generate pDC via lo-hi stage, the other to generate cDC via Zbtb46⁺ Ly6D⁺ stage. However, this has not really been directly shown in this study.

To 43: We had written in the manuscript “that these precursors still contribute to cDC generation via a Zbtb46⁺ Ly6D⁺ transitional state”, which is shown in our data. The results of CellTrace Blue labeling experiments that we added to the revised paper indicate that the Ly6D^{hi} Zbtb46⁻ lo-lo precursor fraction contains cDC-primed cells in

addition to pDC-precursors, which generate cells with cDC phenotype after cell division. See answer to point 10.

44. Line 126: missing reference

To 44: The reference was inserted.

45. Figure 1A. The gating for the preDC is not convincing. The gating on the Sirp-a is weak.

To 45: A better example is shown in revised Fig. 1.

46. Line 158 : Why the authors refer to the S1 and S2. They have not relation within the text.

To 46: Thank you for noting this. It was corrected in the revised text.

47. Fig 1D: the lo-lo population are few transcripts for irf5 and notch4 but present in the pDC and other subsets. Irf5 is associated with IFN response in response to viral infection. Didn't find anything relevant about notch4 but I found that interesting.

48. Line 183 : where does that claim come from? The Fig E do not allow to see gene signatures.

To 48: The gene expression in the sorted populations was compared to published lists of genes that are known to be differentially expressed by pre-cDC, cDC, CDP, pDC and pre-pDC (gene signatures). Distance from the center of the radar plots indicates that the population is enriched in genes present in these gene signatures. This is expressed more clearly in the revised manuscript (p. 6, line 187-189, and legend to Fig. 1d).

49. Line 185 : the author claim is strong. Because it is just based on transcriptome analysis, the utilization of the term low proliferative potential is not justified.

To 49: Thank you for pointing this out. The term was omitted here and replaced by: "with lower expression of genes indicating proliferation compared to pre-cDC and lo lo cells" (p. 6, line 190-191).

50. Fig2A-B : A representative facs plot for the lo-hi populations wouldill be appreciated.

To 50: The representative plot was included in revised Fig. 1e.

51. Also the authors claim line 203 "These results indicate that CD11c+ Siglec-H+ CCR9lo BM cells expressing high levels of B220 preferentially give rise to CCR9hi pDCs while those expressing low levels of B220 generate pDCs and cDCs. CouldDoes the lo-lo population may be contaminated with lo-hi cells I.e B220+ which are described to differentiate principally into pDC?"

To 51: The lo-lo cells were sorted as lo-lo and the purity check after sort indicated that they had high purity. Since lo-hi cells barely proliferate during the 3 day culture

(revised Fig. 4) and pDCs generated from lo-hi cells showed a very low percentage of cells that had diluted the CTB dye after the 3 days of culture are 5 days after in vivo transfer, a minor contamination would not explain the generation of a substantial number of pDCs from lo-lo cells in vitro and in vivo (shown in revised Fig. 4 and 5).

52. Fig D. Maybe it is just me, We I found that hard to compare the wt and IFNR^{-/-} as they are not on the same graph..

53. Fig S2 : a TLR9^{-/-} may have been better to claim that ?

To 52 and 53: The WT vs IFNARKO data is now shown in revised Fig. S9. Since we have 3 conditions (Flt3L alone, Flt3L plus CpG, Flt3L plus IFN- α) and 4 populations we separated the graphs showing cDCs and pDCs output to avoid too many conditions in one graph.

54. Line 223 . Ref missing

55. Line 225 and 230 : error in the figures cited. It should be Fig 2E,G

To 54, 55: Thank you, the errors were corrected.

56. Fig 2.E. Few cDC1 are generated in medium only. Also a representative facs plot for the CpG-A or at least type I-IFN would have been nice.

57. Line 231 : The authors claim "Thus, exposure of CD11c+Siglec-H+CCR9lo precursors to type I IFN limited cDC output in favor of pDC output, but promoted differentiation of cDC1 and increased expression of CD86 and MHC-II". However, the panel G show CD80 and CD86 expression on cDCs derived from lo-lo precursors. The authors should rewritte that conclusion or show the related results.

To 56 and 57: For sake of clarity and more focus, we omitted this part as non-essential.

58. Line 240. Missing ref.

59. Line 268. Missing ref.

60. Line 268. Missing ref.

61. Line 645. Is Ly6D really used in the Lineage?

62. Line 670. Typo error.

To 58 – 62: Thank you for pointing these out. The missing references were inserted and the errors were corrected. Ly6G not Ly6D was used as the Lineage marker. This was a typing error.

Reviewer #2 (Remarks to the Author):

In this study, Musumeci et al identified new pDC precursor populations—CD11c(+) Ly6D(+) Siglec-H(+) CCR9(lo) B220(lo) [lo-lo] precursors and more advanced CD11c(+) Ly6D(+) Siglec-H(+) CCR9(lo) B220(hi) [lo-hi] precursors—placed between previously identified CD11c(-) Ly6D(+) Siglec-H(+) precursors (pre-pDCs) and pDCs. Interestingly, lo-lo precursors still retained the potential to give rise to

cDCs, probably via Siglec-H(+) Zbtb46(+) Ly6D(+) intermediary cells under inflammatory condition *in vivo* and under Flt3L-supplemented *in vitro* culture. Overall, this study is well executed, especially bioinformatics for scRNA-seq and FACS. The concept of the plasticity between pDC and cDC cell fate is an important subject. However, I have some concerns about the cell fate assays and the manuscript writing.

Major points)

1. The authors state “In conclusion, our study demonstrates that CD11c(+) Siglec-H(+) Ly6D(hi) Zbtb46⁻(-) CCR9(lo) B220(lo) precursors retain the potential to give rise to pDCs and cDCs under steady state conditions” (line 591). However, the authors showed the generation of cDCs from lo-lo precursors only in EAE model mice and in Flt3L-supplemented *in vitro* cultures. On the other hand, Fig. 6A indicates that the “transitional” Zbtb46(+) Ly6D(+) Siglec-H(+) CD11c(+) cells exist *in vivo* in a steady state. This seems to indicate that pDC precursors such as pre-pDCs and lo-lo precursors differentiate into cDCs in a steady state *in vivo*. Thus, it is important to demonstrate the results of cell transfer experiments, ideally in non-irradiated normal mice, to test whether pre-pDCs and lo-lo precursors generate cDCs, via Siglec-H(+) Ly6D(+) Zbtb46(+) cells, *in vivo*. Also, do the *in vitro* experiments represent inflammatory or steady conditions?

To 1: We would like to thank the reviewer for this valuable suggestion. To address the cell fate of lo-lo and lo-hi precursors in the steady state *in vivo*, adoptive transfers of Ly6D^{hi} Zbtb46⁻ lo-lo and lo-hi precursors into untreated non-irradiated mice were performed and the progeny were analysed by multiparameter flow cytometry in the spleen showing generation of pDCs, cDC2 and few cDC1 as well as some Zbtb46⁺ Ly6D⁺ cells after 5 days. For comparison Zbtb46⁺ Ly6D⁺ intermediary cells (isolated from BM cells) were also transferred and generation of cDCs was observed (data shown in revised Fig. 5, described on p. 16, line 428-448). These results confirm the contribution of Ly6D^{hi} Zbtb46⁻ lo-lo to cDCs via Siglec-H⁺ Zbtb46⁺ Ly6D⁺ cells in the steady state *in vivo*. The data showing cell fate after transfer into mice with EAE (former Fig. 3) were removed for more clarity as requested by the first reviewer.

2. The evidence that a single precursor cell can generate both pDCs and cDCs needs to be shown to draw the authors' conclusion. At least the authors should perform single cell culture experiments *in vitro* for lo-lo precursors (and pre-pDCs).

To 2: We attempted to do the suggested clonal assays, but due to the low number of divisions we couldn't detect enough cells to assess the cell fate of individual precursors. Ly6D^{hi} Zbtb46⁻ lo-lo precursors generated pDCs and cDCs (revised Fig. 4a, b). To investigate whether these precursors have dual pDC/cDC potential or whether they contain pDC- and cDC-primed cells we followed their division and phenotypic changes by Cell Trace Blue labeling that was done before sorting the subsets in order to obtain comparable labeling. As described on p. 14, line 397-406 and shown in revised Fig. 4f upregulation of Zbtb46 and MHC II as well reduced expression of Ly6D and Siglec-H in the undivided fraction after 3 days of culture indicated that a part of the Ly6D^{hi} Zbtb46⁻ lo-lo precursors had started to differentiate into cDCs before the first division and subsequently expanded. Zbtb46 expression emerged early after 20 hrs of culture before cell division (p. 15, line 406-409; revised Fig. 4 g and h). Addition of IFN- α to the culture inhibited cell division and arrested cDC-primed cells in the Zbtb46⁺ Ly6D⁺ state, while at the same time promoting pDC

differentiation (p. 17, line 471-475; revised Fig. 6). We conclude that “cell fate decision before division argues against dual potential and shows that the Ly6D^{hi} Zbtb46⁻ lo-lo precursor fraction contains cDC-primed cells in addition to pDC precursors.” (p.15, 409-411). We included a graphical display of the two models, dual potential vs precommitment in revised Fig. S9 and describe our interpretation of these results in the discussion (p.19, line 494-503).

3. It would be important to show direct evidence of the hierarchy between the precursor populations (pre-pDCs, lo-lo precursors and lo-hi precursors) at least in vitro (ideally in vivo), not only predicting by bioinformatics.

To 3: Thank you for pointing this out. To address this point Ly6D⁺ Siglec-H⁺ LP (DP) were sorted as Lin⁻ CD16/32⁻ CD11c⁻ B220⁻ Ly6C⁻ CD135⁺ CD117^{lo-int} CD115⁻ IL7R⁺ Ly6D⁺ Siglec-H⁺ cells and cultured with Flt3L. After 4 days few but detectable lo-lo precursors, some lo-hi precursors, pDCs, cDCs and Zbtb46⁺ Ly6D⁺ intermediary cells were identified in the progeny (see revised Fig. S8). The kinetic analysis of the Ly6D⁺ Siglec-H⁺ LP culture (shown in Fig. S8) showed that within the CD11c⁺ Siglec-H⁺ Ly6D⁻ Zbtb46⁻ cells the lo-hi cells exceeded pDCs at the early time points and pDC exceeded lo-hi cells at the later time points. The percentage of lo-lo cells within this fraction was not much changed over time indicating that it is a short transitory state. We found no evidence for generation of lo-lo or lo-hi precursors or pDCs from CDPs that had been sorted as Lin⁻ B220⁻ Ly6C⁻ CD135⁺ CD117^{lo-int} CD115⁺ CD11c⁻ MHCII⁻ (revised Fig. S8). These results are described in the revised manuscript on p. 15, line 413-426 and are discussed on p. 21/22, line 548-559).

4. Do cDCs generated from pDC precursors and those derived from CDPs have distinct functions? What would be the biological meaning of the plasticity?

To 4: We discuss this in the revised manuscript (p. 23, line 594-603).

5. The manuscript is very difficult to read partly due to the indefinite description of surface markers for each cell population. For example, lo-lo precursors are indicated as CD11c(+) MHCII(-/lo) Siglec-H(+) CCR9(lo) pDC-like cells, B220(lo) Ly6D(+) Zbtb46(-) cells, CCR9(lo) B220(lo) cells, CD11c(+) Ly6D(+) Siglec-H(+) CCR9(lo) B220(lo) cells, and so on. Also, I recommend shortening the manuscript, especially the bioinformatics part, to focus on the essential points.

To 5: Thank you for pointing this out. We tried to improve this and use the same description throughout the manuscript. Specific markers are only mentioned when it is important for the understanding of the results. Since the sorting of the lo-lo cells was refined based on our results from scRNA-seq and high-dimensional flow cytometry, we describe the cells as Ly6D^{hi} Zbtb46⁻ lo-lo and lo-hi cells when the refined gating strategy was used (revised Fig. 3, 4, 5, 6). The bioinformatics part of the manuscript has also been shortened as suggested omitting the RACE-ID analysis to avoid redundancy (p. 8-11, line 242-304).

Minor points)

6. Lines 196 to 203: The statements “the lo-lo precursors generated mainly cDCs and few CCR9(hi) pDCs” and “those expressing low levels of B220 generate pDCs and cDCs” appear to be conflicting.

7. Lines 225, 229, and 230: Fig numbers are incorrect (should be Fig. 2).

8. Fig. S4b: Are the lo-lo cells in this Fig a Zbtb46(-) cell fraction as stated in the main text (line 264)? If so, please indicate so in the Fig. This also applies to Fig. 6.

9. Line 276: Are Ly6D(+) LP a Siglec-H(+) fraction of LPs? If so, please indicate so in the Fig. This also applies to Fig. 6. It would be better to state the definition of LPs in the text.

To 7-9: Thank you for pointing out these errors. They were corrected and clarified in the revised manuscript.

10. Fig. 5: Where are lo-lo precursors projected in this RNA velocity assay?

To 10: Projection of the sorted populations is now shown in revised Fig. 2a of the revised manuscript.

11. Fig. 6A: "/" within "Lin(-) Flt3(-) CD11c(+) B220(-) Siglec-H(-)/p1" is unclear. Does this "/" mean "and"? Because "Lin(-) Flt3(-) CD11c(+) B220(-) Siglec-H(-)" is identical to the p1 in Fig. 1, it is confusing. Does "Siglec-H(-)/(+) pre-cDC" mean "Siglec-H(-) or Siglec-H(+) pre-cDC" ?

To 11: To make this easier to understand, we have included the complete gating strategy in revised Fig. 3a and indicated subsequent gating with arrows.

Reviewer #3 (Remarks to the Author):

The main claim of the paper is that there is plasticity between the lineages of plasmacytoid dendritic cells (pDCs) and conventional dendritic cells (cDCs). Specifically, the author identified a bone marrow (BM) precursor stage coined "lo-lo" that is already engaged in differentiation towards pDCs but can still give rise to cDCs, and whose fate is influenced by the inflammatory milieu since type I interferon (IFN-I) promoted its differentiation towards pDCs over cDCs in vitro. This proximal BM precursor of pDCs is phenotypically characterized as Lineage(CD3, CD19, Ly6G, NK1.1, Ter119)-neg CD135-high CD11c-pos Siglec-H-pos CCR9-low B220-low Ly6D-high Zbtb46-neg. It is also Ly6C-neg, CD115-pos and CXCR4-pos.

Moreover, by combining single cell RNA-sequencing (scRNA-seq), high content flow cytometry, and in vitro differentiation assays with kinetic measurement of key lineage-marker genes by qRT-PCR, the authors mapped the differentiation trajectory of this "lo-lo" precursor into pDCs and cDCs. They showed that this "lo-lo" precursor gave rise to cDCs by going through a transient state co-expressing Ly6D and Zbtb46. Immediately following ex vivo purification from BM, this Ly6D-pos Zbtb46-pos transient state expressed key pDC-lineage marker genes including Tcf4, Spib and Irf8. However, it lost the expression of these pDC-lineage marker genes under the differentiation conditions used by the authors in vitro while inducing key cDC-lineage marker genes including Batf3, Id2 and Spi1.

The "lo-lo" precursor characterized and studied by the authors expresses CD11c. Hence, it is different from the BM precursor shared between pDCs and B cells that

has been previously identified as also Siglec-H-pos Ly6D-high but CD11c-neg (“Ly6D-pos lymphoid progenitor (LP)”, manuscript references 37 & 38). Most importantly, the authors convincingly showed that the “lo-lo” progenitor gave rise to both pDCs and cDCs, contrary to the “Ly6D-pos LP” that was reported not to yield cDCs but exclusively B cells and pDCs (ref. 37 & 38).

This study is original and convincing. The results support rather well the conclusions drawn. It should be of great interest for the researchers interested in the ontogeny and functions of pDCs and cDCs, since it brings novel and important information regarding the plasticity of these two lineages of immune cells and their relationships. This is timely considering the ongoing controversy on the identity of pDCs, which have been proposed recently not to belong to the family of DCs but to that of innate lymphoid cells (ref. 37). The data in this manuscript show that proximal pDC progenitors are plastic and can differentiate into cDC, showing that there exist an ontogeny proximity between pDC and cDC. Importantly, these results are consistent with the reciprocal molecular regulation of the differentiation of pDCs versus type 1 cDCs by opposing combinations of a handful of master transcription factors whose artificial perturbation can lead to the reprogramming of pDCs into cells resembling cDC1s or conversely. This is discussed well by the authors (lines 74-78, and 525-531, manuscript references 31-35, 61, 62). Hence, this study should become a key component of the scientific corpus advancing our understanding of pDC identity and of their relationship to cDCs.

Two major points and a few minor points need to be addressed.
Major points.

1) Alternative explanations for the effects of IFN-I on the output of in vitro differentiation cultures.

The output of in vitro cultures are given as percent of pDCs or cDCs within CD11c-pos cells (Figures 2c-d, 2f; 6c-d). Neither absolute numbers of DC subsets nor global cell yields of these cultures are given. This is a major issue, because IFN-I have anti-proliferative and pro-apoptotic functions differentially affecting distinct cell types. Hence, IFN-I could induce the preferential apoptosis of the cells engaged in cDC differentiation, and/or inhibit their proliferation, while sparing the cells engaged into pDC differentiation. Such a mechanism of differential selection acting on a population of heterogeneous precursors already committed to either the pDC or cDC lineage could explain the strong relative enrichment observed in the output of pDCs in the cultures. Hence, one cannot exclude alternative explanations to the interpretation given by the authors that IFN-I instructs the fate choice of individual bipotent BM precursors towards pDCs over cDCs. The authors must address this issue experimentally, by providing absolute numbers of cDCs and pDCs retrieved per well for each culture condition, or at least the total yield of cells per well. For each input precursor population, they could also measure cell apoptosis and proliferation by flow cytometry kinetically over time comparing medium alone to IFN-I.

To 1) We thank the reviewer for raising this important point. We had observed that IFN- α treatment of the lo-lo precursors increased the relative output of pDCs while inhibiting the relative output of cDCs. We took up the reviewer’s suggestion and repeated the experiments investigating the effect of IFN- α on cDC vs pDC output comparing 6 populations that were isolated using the refined gating strategy that is

shown in revised Fig. 3a. CellTrace Blue labeling was performed before sorting to allow for comparison of CTB dilution in the progeny of the different input populations. The results are shown in revised Fig. 6 and Fig. S9 and described on p.16-18, line 450-480:

Addition of IFN- α increased the percentage and absolute number of pDCs and decreased the percentage and absolute number of cDCs generated from lo-lo precursors within 3 days of culture. Interestingly, we found that the percentage and number of Zbtb46⁺ Ly6D⁺ cells tended to be increased indicating that IFN- α arrested cDC-primed cells in the Zbtb46⁺ Ly6D⁺ intermediary state (revised Fig. 6a, b, c, d and Fig. S9). Congruence of percentages (revised Fig. 6a, b, c) and absolute numbers (revised Fig. S9e, g, i) of the generated cell types showed that the reduced cDC output is not due to increased apoptosis of cDCs or cDC-primed cells. Instead, we found that the cells engaged in cDC differentiation (Zbtb46⁺) failed to expand in cultures containing IFN- α (revised Fig. 6 e) Additionally, IFN- α increased the percentage of CCR9^{high} B220^{high} pDCs within the CD11c⁺ Siglec-H⁺ Ly6D⁺ Zbtb46⁻ fraction of the progeny (revised Fig. 6f, g). We conclude that IFN- α had a selective rather than instructive effect on pDC vs cDC output from the lo-lo precursors, but additionally promoted full differentiation of pDCs. This is described and discussed in the revised manuscript (p. 17/18, line 477-478 and p. 24, line 610-613).

2) In vivo fate plasticity of the Ly6D-high Zbtb46-neg “lo-lo” precursors.

2-1) The experiment of the in vivo cell fate of transferred “lo-lo” precursors shown in figure 3 is not conclusive, because this population is heterogeneous and encompasses a significant fraction of Ly6C-pos Zbtb46-pos pre-cDCs, as shown later in the manuscript. Hence, to rigorously demonstrate that the “lo-lo” precursor cells can give rise to both pDCs and cDCs in vivo, the authors need to repeat that experiment using the strategy designed later in the paper to remove the contaminating pre-cDCs, namely sorting Ly6D-high Zbtb46-neg “lo-lo” precursors.

To 2-1) As suggested by the reviewers we have performed transfer experiments using the Ly6D^{hi} Zbtb46⁻ lo-lo and lo-hi precursors (sort gates as in revised Fig. 3a) into untreated and non-irradiated congenic mice. The progeny were analysed by multiparameter flow cytometry in the spleen showing generation of pDCs, cDC2 and few cDC1 as well as some Zbtb46⁺ Ly6D⁺ cells after 5 days. Lo-hi cells preferentially generated pDCs *in vivo* in line with them being an immediate precursor of pDCs. For comparison Zbtb46⁺ Ly6D⁺ intermediary cells (isolated from BM cells) were also transferred and generation of cDCs was observed after 3 days (shown in revised Fig. 5, described on p. 16, line 428-448).

2-2) The data from figure 3 show that, upon in vivo adoptive transfer, the “lo-lo” precursors gave preferentially rise to pDCs (~55% of CD45.1-pos cells) over cDCs (~30%). This is strikingly different from what was observed with the in vitro differentiation assay, as shown on figure 2c-d where the output was on the contrary strongly biased towards cDCs (~80%) over pDCs (~10%). It is not clear whether this discrepancy reflects major differences in the differentiation process as it occurs in vivo as opposed to in vitro, or whether it is due to the inflammatory condition associated to the EAE model. Hence, the authors must compare the in vivo differentiation of the Ly6D-high Zbtb46-neg “lo-lo” precursors upon their adoptive transfer not only in animals induced to develop EAE but also in control animals at ground state (no inflammation). Ideally, the authors should also include recipient

animals injected by CpG or infected with a virus, with adoptive transfer of the Ly6D-high Zbtb46-neg “lo-lo” precursors at the time of the peak production of IFN-I in vivo, to better mirror the inflammation conditions used in vitro.

To 2-2) As suggested we have omitted the *in vivo* cell fate data in mice with ongoing EAE and have performed the recommended transfer experiment in steady state mice described above (shown in revised Fig. 5 of the revised paper). We decided to focus on the non-inflammatory condition for sake of clarity and conciseness as recommended by several reviewers and therefore did not further pursue cell transfer experiments in mice injected with CpG.

Minor points.

3) Please give additional information to allow the reader comparing more precisely the complementary analyses of the scRNA-seq data from Figure 4 & 5. Figure 4 & 5 correspond to different, complementary, analyses of the same scRNA-seq dataset, and the clustering of cells were performed independently with distinct methods for each analysis (RaceID yielding 8 clusters in Figure 4, versus Louvain from scvelo yielding 7 clusters in Figure 5). It is not clear if the cells identified as monocytes in Figure 4 (cluster 7) are still included in the analysis of Figure 5, and if yes, to which cluster do they belong. Does cluster 9 from Figure 4 correspond to Ly6D-pos LP and hence to cluster 4 from Figure 5? Does cluster 4 from Figure 4 correspond to cluster 3 from Figure 5? To help the reader comparing these analyses, the authors should provide a table/heatmap indicating the % overlap in the individual cell content of each (Figure 4/Figure 5) cluster pair.

To 3) To shorten and focus the manuscript as suggested by the reviewers we have removed the RACE ID analysis results and the corresponding results text and now only show the results of the scvelo analysis including projection of the sorted cells onto the diffusion map (revised Fig. 2a).

4) Please discuss the discrepancy between your study and those on the Ly6D-pos Siglec-H-pos LP.

As the authors proposed, it is likely that the CD11c-pos “lo-lo” precursor they characterized is just downstream of the CD11c-neg Ly6D-pos Siglec-H-pos LP previously identified and characterized by others (ref. 37 & 38). In that case, the fact that the CD11c-neg Ly6D-pos Siglec-H-pos LP was reported to give rise only to B cells and pDCs but not to cDCs is inconsistent with the authors’ results showing that the CD11c-pos “lo-lo” precursor gave rise to both pDCs and cDCs. Could the authors please discuss this discrepancy?

To 4) Thank you for pointing this out. We discuss this in more detail now in the revised manuscript (p. 21, line 552-556). In fact, in the report by Rodrigues et al. it can be seen that a small percentage of Ly6D⁺ Siglec-H⁺ LP (called DP) also gave rise to cDC (10-20%, shown in the supplement and not discussed). We observed that Ly6D⁺ Siglec H⁺ LPs sorted as described in that paper and cultured with Flt3L gave rise to lo-lo, lo-hi cells and pDCs as well as to Zbtb46⁺ Ly6D⁺ intermediate cells and cDCs (revised Fig. S8). We also observed that the output of pDCs and cDCs was dependent on the time point of the analysis with higher frequency of lo-hi cells and pDCs observed at earlier timepoints and cDCs accumulating at later time points (revised Fig. S8). The fact that we see a higher output of cDCs from these cells than

described in the Rodrigues et al. paper could be due to specifics of cell isolation and culture conditions (medium, serum, Flt3L source) or even housing conditions of the mice from which the cells were isolated.

5) Please italicize the names of genes throughout the paper, both in the text and on the figures.

6) Please carefully revise and correct figure and table calling in the main text and in figure legends.

There are mistakes in figure and table calling in the main text and in figure legends. A non-exhaustive list of example is provided thereafter. In the legend of figure 5 panel a, the text "(same data as in Fig. 5)" must be corrected as "(same data as in Fig. 4)". Line 225, "(Fig. 1E, G)" must be replaced by "(Fig. 2E, G)". Line 229, "(Fig. 1F)" must be replaced by "(Fig. 2F)". Line 230, "(Fig. 1G)" must be replaced by "(Fig. 2G)". Line 645, replace "(see Supplemental Table 1)" by "(see Table S3)", etc...

7) Line 645, there is a mistake in the list of antibodies used in the lineage-depletion cocktail: replace "Ly6D" by "Ly6G".

To 5-7: Thank you for pointing out these errors, which we have corrected in the revised manuscript.

8) Literature citation.

The authors have covered the relevant literature accurately and rather extensively. However, when mentioning the specific finding that "DC subpopulations are conserved across species" line 50, they should cite the first result papers that demonstrated it for the first time (PMID: 18218067; 20193019). Alternately, they could cite a relatively recent review specifically discussing this concept, from authors who put it forward by comparing mouse and human DCs and then extended it across several warm blooded vertebrate species (e.g. PMID: 26082777).

To 8: Thank you for noting this. We have cited the original paper showing the species conservation of DC subsets in the revised manuscript (p. 1, line 60).

REVIEWERS' COMMENTS

Reviewer #1 (Remarks to the Author):

Musumeci and Colleagues have made a valiant attempt at recrafting their manuscript to make it more coherent, and have addressed every issue we raised. They should be commended for what has clearly been quite a lot of work so that, now, it is clear exactly where their progenitor fits into prior knowledge in the field, and what the phenotypic, molecular and cell fate characteristics of their population are. The revised version is clear, convincing and will be an important contribution to the field.

We have some additional points that require attention
Although we could see the data in the rebuttal that constitutes Supplementary Fig 8, the Figure itself is missing in the compiled document we received.

Please double check that every mention of a Figure in Results and Discussion actually cites the Figure/panel number.

The authors discuss two models to explain the differentiation into cDCs/pDCs from the Zbtb46- lo-lo precursors with Model A, dual pDC/cDCs potential in individual lo-lo precursors, and Model B, suggesting an heterogeneity among the lo-lo precursors. The author's results support the Model B, and we agree that the Model A is less probable. To better highlight the heterogeneity of the lo-lo precursors' population, a clearer annotation of the transcriptomic analysis (Figure 2e and Supl.4.) would be useful. The relation between the Louvain clusters and the cell types (e.g. Lo-lo etc) should be more explicit in the figure to complement the corresponding results section. In Figure 2a, the lo-lo population seems to be spread among most of the other Louvain clusters in terms of transcriptomic signature, and a bi-modal expression of Zbtb46 is visible (even if excluded in the rest of the paper), but it is hard to position them among the Louvain Clusters (Fig2e. and Supl.4) without reading the results section. This information will highlight if the lo-lo population is shared among specific Louvain clusters, arguing with their heterogeneity. In addition, the expression of Zbtb46 on the velocity plot Fig.2g would be nice, to determine if the Zbtb46hi cluster closely with the CDP and see how heterogeneous the Zbtb46 - lo-lo precursors are.

Reviewer #2 (Remarks to the Author):

Major comments

The authors have tried to address essentially all of my comments by performing a number of experiments. I really appreciate their hard work. However, the obtained results suggested that the lo-lo population is a mixture of pDC precursors and cDC-primed cells, rather than harboring pDC-cDC bipotential cells. Given the many reports demonstrating the specification of cell fate at much earlier stages when analyzed at a single cell level as opposed to at a bulk population level, the authors' new data are not surprising: it is still possible that cDCs are generated from lymphoid progenitors via the lo-lo precursors. In fact, the authors now show that Ly6D+ Siglec-H+ lymphoid progenitors can generate cDCs in vitro, although it is unclear whether this occurred via the lo-lo cells, because only few lo-lo cells were detected in their assay. Overall, I regret to mention that this paper has not answered their own question, i.e., the separation of pDC and cDC lineages. For example, the statement at lines 604-605 "the contribution to cDCs from pDC-primed lymphoid-derived precursors" is not convincing. The authors could either perform clonal experiments using progenitors at much earlier stages or restructure the paper by stating only what they can conclude.

Minor comments

1. The manuscript is still hard to read. I recommend that the authors write more "interpretation" of the results rather than simply describing the data in detail, even in the Results section. The Abstract would be difficult to understand unless the readers read the entire manuscript. For example, what "this compartment" on line 43 means is unclear.
2. Supplementary Fig 7d: Is the lo-lo fraction completely absent on day 0?
3. Line 229: 10.000 should be 10,000
4. Supplementary Fig 9e: 50.000 should be 50,000

Reviewer #3 (Remarks to the Author):

To the best of my appreciation, the authors have satisfactorily answered all of the reviewers' requests.

The attempts at in vitro clonal differentiation assays were not successful, due to the low output of the advanced precursors that the authors are characterizing. However, the authors have generated a wealth of new data. (1) They completed and better exploited their characterization of the phenotype of the precursors from bone marrow, through high content spectral flow cytometry, including by comparing them to cell populations isolated from the spleen and encompassing the pDC-like cells and tDCs recently described by others (new Figure 3). They performed in vivo transfer of purified progenitor populations to track simultaneously their proliferation and their differentiation (new Figure 5). These experiments included an unbiased comparison of the phenotype of the cells differentiated upon in vivo transfer experiments of purified precursor populations from donor CD45.2 animals, in comparison with endogenous splenic cell populations from the recipient CD45.1 mice, by high content spectral flow cytometry (Figure 5 c-d). They also followed cell death and proliferation during their study of the impact of the exposure of precursors to IFN-I during in vitro bulk differentiation assays (new Figure 6, in particular panels c and e). These experiments confirmed the major conclusions from the initial study and helped refining some interpretation.

The demonstration is original, robust and interesting, that the lo-lo precursors generate both pDC and cDC, not only in vitro but also in vivo, and that their differentiation into cDC occurs through a transient Ly6D+ Zbtb46+ stage. This discovery has major implication for better understanding the ontogenetic and functional relationships between pDC and cDC, in the frame of the ongoing controversy regarding the lymphoid versus dendritic nature of pDC.

In the revised manuscript, the hypothesis favored regarding the mechanisms underlying is that this cell population is heterogeneous, encompassing a mix of pDC-primed and cDC-primed cells (Figure S9k – B), rather than being a homogeneous population of bi-potent cells (Figure S9k – A). As far as I understand, this interpretation is opposite to the one put forward in the initial version of the study. However, it is indeed supported by the novel data generated and shown: the kinetic analysis of the in vitro differentiation and proliferation of the lo-lo precursors (new Figure 4 e-h), and the demonstration that IFN-I effects promoting pDC over cDC differentiation from this precursor population occur through selective rather than instructive effects (new Figure 6).

Following reviewer#1 excellent advice, the authors have extensively re-organized and shortened the result section, which succeeded in greatly improving the reasoning flow and readability of the story. They also extensively edited the discussion section, accordingly to their new results and the resulting changes in some interpretation.

Thus, the manuscript has been greatly improved. I do not have any major point anymore. A few

typographical or stylistic minor issues remain that could be addressed at the time of production for publication, provided that the other reviewers and the editors also find the present manuscript suitable for publication in Nature Communications.

Minor points.

- 1) Supplementary Figure 8 is numbered as Supplementary Figure 7 in its caption (Supplemental Material file).
- 2) A scale is lacking on Figure 1d (i.e. indicate for each circle shown the % shared genes between the DEGs identified by bulk RNA-seq between sorted populations and the reference cell type-specific transcriptomic signatures).
- 3) Gene names must be italicized on Figure 2d (heatmap), Figure 4d (heatmap), Supplementary Figure 2 (heatmap), and Supplementary Figure 4 (on the heatmap and for the title of the Velocity graphs).
- 4) Some gene names are cut, on the left of the heatmap in Supplementary Figure 4c (e.g. Tmem108, ...7H13Rik, ...45G16.5, ...4K13Rik).
- 5) The gene markers for the Louvain clusters are shown on Figure 1d before the clusters themselves are shown (Figure 2e). Maybe Figure 2a could be adapted by using a mix of different colors for the sorting phenotype of the cells as it is currently the case, combined with different symbols/shapes for the Louvain clusters?
- 6) Supplementary Figure 4c is cited before Figures 4a and 4b, in the main text. Reorder and renumber the panels accordingly to their citation order in the main text?
- 7) Line 276, the gene name is written "Kif1" as opposed to "Kif11" in Supplementray Figure 2b.
- 8) Line 401, in "this raises the question, if", I think that "if" should be replaced by "whether".
- 9) Lines 402-404, add a coma between "culture" and "cell division"; add "the" between "occurred in" and "progeny".
- 10) Line 407, add a hyphen between "Siglec" and "H".
- 11) Line 437, add a coma between "steady state" and "these populations were sorted".
- 12) Line 447, add a coma between "not pDCs" and "showed evidence".
- 13) Line 506, add a coma between "stimulation" and "cell engaged in cDC".
- 14) Lines 605-610, same point repeated 2x, on cDC2 ontogeny: remove end "and the first wave ... resident LP4".
- 15) Line 610, I think that "If" should be replaced by "Whether".
- 16) Line 629, add a coma between "precursors" and "indicating".
- 17) Line 685, add a coma between "LP39" and "the lineage markers".
- 18) Lines 1246-1248, please rewrite "cDCs generated from each precursor subset and for comparison pDCs generated from lo-hi precursors were manually gated as CD11c+ Zbtb46high MHCIIhigh and projected onto the UMAP." as "cDCs generated from each precursor subset were manually gated as CD11c+ Zbtb46high MHCIIhigh and projected onto the UMAP , together with pDCs generated from lo-hi precursors for comparison."
- 19) Line 1304, I think that "Zbtb46(neg)" should be added after Ly6D+, in the sentence "Percentage of differentiated pDCs within CD11c+ Siglec-H+ Ly6D+ cells derived from lo-lo and lo-hi precursors after 3d of culture with and without IFN-alpha addition (n=5)".

Point-by point Reply to the reviewers' comments

We thank the reviewers for their feedback and reply to the individual comments below.

Reviewer #1 (Remarks to the Author):

Musumeci and Colleagues have made a valiant attempt at recrafting their manuscript to make it more coherent, and have addressed every issue we raised. They should be commended for what has clearly been quite a lot of work so that, now, it is clear exactly where their progenitor fits into prior knowledge in the field, and what the phenotypic, molecular and cell fate characteristics of their population are. The revised version is clear, convincing and will be an important contribution to the field.

We have some additional points that require attention

Although we could see the data in the rebuttal that constitutes Supplementary Fig 8, the Figure itself is missing in the compiled document we received.

Thank you for the thorough review of our revised manuscript.

Supplementary Figure 8 was accidentally labeled as Supplementary Figure 7. Thank you for noticing this. It was corrected.

Please double check that every mention of a Figure in Results and Discussion actually cites the Figure/panel number.

This was checked.

The authors discuss two models to explain the differentiation into cDCs/pDCs from the Zbtb46- lo-lo precursors with Model A, dual pDC/cDCs potential in individual lo-lo precursors, and Model B, suggesting an heterogeneity among the lo-lo precursors. The author's results support the Model B, and we agree that the Model A is less probable. To better highlight the heterogeneity of the lo-lo precursors' population, a clearer annotation of the transcriptomic analysis (Figure 2e and Supl.4.) would be useful. The relation between the Louvain clusters and the cell types (e.g. Lo-lo etc) should be more explicit in the figure to complement the corresponding results section. In Figure 2a, the lo-lo population seems to be spread among most of the other Louvain clusters in terms of transcriptomic signature, and a bi-modal expression of Zbtb46 is visible (even if excluded in the rest of the paper), but it is hard to position them among the Louvain Clusters (Fig2e. and Supl.4) without reading the results section. This information will highlight if the lo-lo population is shared among specific Louvain clusters, arguing with their heterogeneity.

Thank you for pointing this out, we have adapted Figure 2 to make this clearer. In the diffusion map in Figure 2a the brown symbols showing the cells sorted as lo-lo cells are now in the foreground for better visibility. In addition, we moved the former Figure 2e and f (showing the Louvain clusters in different colours) above the heatmap, so that they can be viewed directly below the Figure 2a for direct visual comparison allowing allocation of the sorted cell identity to the Louvain clusters. The map showing Ly6D expression as colour overlay was also enlarged for better visibility. It can be seen that the lo-lo cells which cluster with pre-cDCs show low Ly6D expression, whereas the lo-lo cells which cluster with Siglec H⁺ Ly6D⁺ LP show high Ly6D expression.

In addition, the expression of Zbtb46 on the velocity plot Fig.2g would be nice, to determine if the Zbtb46^{hi} cluster closely with the CDP and see how heterogeneous the Zbtb46⁻ lo-lo precursors are.

Zbtb46 transcripts were not detected in the scRNAseq data set and Zbtb46 expression could therefore not be shown as colour overlay. In the results text we refer to the expression of other pre-cDC genes including Id2 and Batf3 in cells sorted as CDP, pre-cDCs and a subset of cells sorted as lo-lo cells (shown in Supplementary Figure 4). This is in line with flow cytometric detection of the Zbtb46-eGFP signal in a fraction of the lo-lo cells when they were gated as CD11c⁺ Siglec H⁺ CCR9^{lo} B220^{lo} (Figure 2c). As described in the results text these results led to our refined gating strategy excluding any Zbtb46⁺ Ly6D⁻ pre-cDCs before gating the lo-lo, lo-hi precursors and pDCs.

Reviewer #2 (Remarks to the Author):

Major comments

The authors have tried to address essentially all of my comments by performing a number of experiments. I really appreciate their hard work. However, the obtained results suggested that the lo-lo population is a mixture of pDC precursors and cDC-primed cells, rather than harboring pDC-cDC bipotential cells. Given the many reports demonstrating the specification of cell fate at much earlier stages when analyzed at a single cell level as opposed to at a bulk population level, the authors' new data are not surprising: it is still possible that cDCs are generated from lymphoid progenitors via the lo-lo precursors. In fact, the authors now show that Ly6D⁺ Siglec-H⁺ lymphoid progenitors can generate cDCs in vitro, although it is unclear whether this occurred via the lo-lo cells, because only few lo-lo cells were detected in their assay. Overall, I regret to mention that this paper has not answered their own question, i.e., the separation of pDC and cDC lineages. For example, the statement at lines 604-605 "the contribution to cDCs from pDC-primed lymphoid-derived precursors" is not convincing. The authors could either perform clonal experiments using progenitors at much earlier stages or restructure the paper by stating only what they can conclude.

Thank you for the thorough review of our revised manuscript.

We agree with the reviewer that we cannot draw conclusions regarding a putative proximal clonal progenitor of pDCs and cDCs. In the introduction (line 201) "... and the question remains in which precursor cells it [separation of pDCs and cDCs] may occur" was removed and the part "Given the known cross-regulation ... we elucidated the heterogeneity and commitment of DC precursor subsets" was replaced by "Here we resolve the heterogeneity and commitment of DC precursor subsets using single-cell RNA sequencing and trajectory inference from RNA velocity combined with cell fate analysis" (lines 227 - 229).

To account for the possibility of an earlier clonal progenitor of pDCs and cDCs we added to the discussion (lines 680 - 691): "The pDC- and cDC-primed cells within the Ly6D^{hi} Siglec-H⁺ Zbtb46⁻ lo-lo precursor fraction may originate from an earlier common clonal progenitor or from separate progenitors which have a similar phenotype and gene expression profile at this stage."

This text passage in the discussion was removed to avoid any overinterpretation (see line 777): “In which situations the contribution to cDCs ... cDC development from CDP and pre-cDC is impaired.”

Minor comments

1. The manuscript is still hard to read. I recommend that the authors write more “interpretation” of the results rather than simply describing the data in detail, even in the Results section. The Abstract would be difficult to understand unless the readers read the entire manuscript. For example, what “this compartment” on line 43 means is unclear.

We included some interpretative sentences in the results section as suggested, but our interpretation of the data is mainly described in the discussion. The abstract was revised for more clarity.

2. Supplementary Fig 7d: Is the lo-lo fraction completely absent on day 0?
Yes, as the lo-lo cells were sorted CD11c-negative on d0.

3. Line 229: 10.000 should be 10,000

4. Supplementary Fig 9e: 50.000 should be 50,000
All the corrections were done as suggested.

Reviewer #3 (Remarks to the Author):

To the best of my appreciation, the authors have satisfactorily answered all of the reviewers' requests.

The attempts at in vitro clonal differentiation assays were not successful, due to the low output of the advanced precursors that the authors are characterizing. However, the authors have generated a wealth of new data. (1) They completed and better exploited their characterization of the phenotype of the precursors from bone marrow, through high content spectral flow cytometry, including by comparing them to cell populations isolated from the spleen and encompassing the pDC-like cells and tDCs recently described by others (new Figure 3). They performed in vivo transfer of purified progenitor populations to track simultaneously their proliferation and their differentiation (new Figure 5). These experiments included an unbiased comparison of the phenotype of the cells differentiated upon in vivo transfer experiments of purified precursor populations from donor CD45.2 animals, in comparison with endogenous splenic cell populations from the recipient CD45.1 mice, by high content spectral flow cytometry (Figure 5 c-d). They also followed cell death and proliferation during their study of the impact of the exposure of precursors to IFN-I during in vitro bulk differentiation assays (new Figure 6, in particular panels c and e). These experiments confirmed the major conclusions from the initial study and helped refining some interpretation.

The demonstration is original, robust and interesting, that the lo-lo precursors generate both pDC and cDC, not only in vitro but also in vivo, and that their differentiation into cDC occurs through a transient Ly6D+ Zbtb46+ stage. This discovery has major implication for better understanding the ontogenetic and functional relationships between pDC and cDC, in the frame of the ongoing

controversy regarding the lymphoid versus dendritic nature of pDC.

In the revised manuscript, the hypothesis favored regarding the mechanisms underlying is that this cell population is heterogeneous, encompassing a mix of pDC-primed and cDC-primed cells (Figure S9k – B), rather than being a homogeneous population of bi-potent cells (Figure S9k – A). As far as I understand, this interpretation is opposite to the one put forward in the initial version of the study. However, it is indeed supported by the novel data generated and shown: the kinetic analysis of the in vitro differentiation and proliferation of the lo-lo precursors (new Figure 4 e-h), and the demonstration that IFN-I effects promoting pDC over cDC differentiation from this precursor population occur through selective rather than instructive effects (new Figure 6).

Following reviewer#1 excellent advice, the authors have extensively re-organized and shortened the result section, which succeeded in greatly improving the reasoning flow and readability of the story. They also extensively edited the discussion section, accordingly to their new results and the resulting changes in some interpretation.

Thus, the manuscript has been greatly improved. I do not have any major point anymore. A few typographical or stylistic minor issues remain that could be addressed at the time of production for publication, provided that the other reviewers and the editors also find the present manuscript suitable for publication in Nature Communications.

Thank you for the thorough review of our revised manuscript.

Minor points.

1) Supplementary Figure 8 is numbered as Supplementary Figure 7 in its caption (Supplemental Material file).

2) A scale is lacking on Figure 1d (i.e. indicate for each circle shown the % shared genes between the DEGs identified by bulk RNA-seq between sorted populations and the reference cell type-specific transcriptomic signatures).

3) Gene names must be italicized on Figure 2d (heatmap), Figure 4d (heatmap), Supplementary Figure 2 (heatmap), and Supplementary Figure 4 (on the heatmap and for the title of the Velocity graphs).

4) Some gene names are cut, on the left of the heatmap in Supplementary Figure 4c (e.g. Tmem108, ...7H13Rik, ...45G16.5, ...4K13Rik).

5) The gene markers for the Louvain clusters are shown on Figure 2d before the clusters themselves are shown (Figure 2e). Maybe Figure 2a could be adapted by using a mix of different colors for the sorting phenotype of the cells as it is currently the case, combined with different symbols/shapes for the Louvain clusters?

The suggested corrections were done. To 5): Former Figure 2e and f (showing the Louvain clusters in different colours) are now shown above the heatmap (former Figure 2d), so that they can be viewed directly below the Figure 2a for direct visual comparison allowing allocation of the sorted cell identity to the Louvain clusters. We couldn't change the shape and colour of the symbols as scvelo functions do not allow for that kind of customization and the images exported from scvelo analysis are not editable to that degree.

- 6) Supplementary Figure 4c is cited before Figures 4a and 4b, in the main text. Reorder and re-number the panels accordingly to their citation order in the main text?
- 7) Line 276, the gene name is written "Kif1" as opposed to "Kif11" in Supplementray Figure 2b.
- 8) Line 401, in "this raises the question, if", I think that "if" should be replaced by "whether".
- 9) Lines 402-404, add a coma between "culture" and "cell division"; add "the" between "occurred in" and "progeny".
- 10) Line 407, add a hyphen between "Siglec" and "H".
- 11) Line 437, add a coma between "steady state" and "these populations were sorted".
- 12) Line 447, add a coma between "not pDCs" and "showed evidence".
- 13) Line 506, add a coma between "stimulation" and "cell engaged in cDC".
- 14) Lines 605-610, same point repeated 2x, on cDC2 ontogeny: remove end "and the first wave ... resident LP4".
- 15) Line 610, I think that "If" should be replaced by "Whether".
- 16) Line 629, add a coma between "precursors" and "indicating".
- 17) Line 685, add a coma between "LP39" and "the lineage markers".
- 18) Lines 1246-1248, please rewrite "cDCs generated from each precursor subset and for comparison pDCs generated from lo-hi precursors were manually gated as CD11c+ Zbtb46high MHCIIhigh and projected onto the UMAP." as "cDCs generated from each precursor subset were manually gated as CD11c+ Zbtb46high MHCIIhigh and projected onto the UMAP , together with pDCs generated from lo-hi precursors for comparison."
- 19) Line 1304, I think that "Zbtb46(neg)" should be added after Ly6D+, in the sentence "Percentage of differentiated pDCs within CD11c+ Siglec-H+ Ly6D+ cells derived from lo-lo and lo-hi precursors after 3d of culture with and without IFN-alpha addition (n=5)"

Thank for the careful review. The manuscript was edited according to these suggestions.